# Regulation of priming effect by soil organic matter stability over a broad geographic scale

Leiyi Chen [1,9], Li Liu[1,2,9], Shuqi Qin[1,2], Guibiao Yang[1,2], Kai Fang[1,2], Biao Zhu [3], Yakov Kuzyakov[4,5,6], Pengdong Chen[1,7], Yunping Xu[8] & Yuanhe Yang [1,2]*

The modification of soil organic matter (SOM) decomposition by plant carbon (C) input (priming effect) represents a critical biogeochemical process that controls soil C dynamics. However, the patterns and drivers of the priming effect remain hidden, especially over broad geographic scales under various climate and soil conditions. By combining systematic field and laboratory analyses based on multiple analytical and statistical approaches, we explore the determinants of priming intensity along a 2200 km grassland transect on the Tibetan Plateau. Our results show that SOM stability characterized by chemical recalcitrance and physico-chemical protection explains more variance in the priming effect than plant, soil and microbial properties. High priming intensity (up to 137% of basal respiration) is associated with complex SOM chemical structures and low mineral-organic associations. The dependence of priming effect on SOM stabilization mechanisms should be considered in Earth System Models to accurately predict soil C dynamics under changing environments.

[1] State Key Laboratory of Vegetation and Environmental Change, Institute of Botany, Chinese Academy of Sciences, Beijing 100093, China. [2] University of Chinese Academy of Sciences, Beijing 100049, China. [3] Institute of Ecology, College of Urban and Environmental Sciences, and Key Laboratory for Earth Surface Processes of the Ministry of Education, Peking University, Beijing 100871, China. [4] Department of Soil Science of Temperate Ecosystems, University of Göttingen, 37077 Göttingen, Germany. [5] Department of Agricultural Soil Science, University of Göttingen, 37077 Göttingen, Germany. [6] Agro-Technological Institute, RUDN University, Moscow 117198, Russia. [7] College of Life Sciences, Ludong University, Yantai 264025, China. [8] Shanghai Engineering Research Center of Hadal Science and Technology, College of Marine Sciences, Shanghai Ocean University, Shanghai 201306, China. [9]These authors contributed equally: Leiyi Chen, Li Liu. *email: yhyang@ibcas.ac.cn

Soil organic matter (SOM) plays a critical role for soil fertility, food production, climate regulation and ecosystem stability[1]. As the largest organic carbon (C) pool in terrestrial ecosystems, the soil C stock is determined by the balance between C inputs from plants and C outputs through microbial decomposition[1] and erosion[2], with these processes occurring interactively in nature[3]. Of the various plant–soil interactions, the priming effect (i.e. alteration of SOM decomposition by labile C inputs) is a key mechanism affecting the soil C cycle[4]. Incorporating the priming effect into Earth System Models can improve the prediction of soil C stocks[5,6]. However, the large variability in the priming effect among various ecosystems ranging from a 380% increase to a 50% reduction in carbon dioxide ($CO_2$) flux from soil[7] greatly impedes the priming representation in Earth System Models. Knowledge concerning the dominant priming driver is thus crucial for model developments to accurately predict soil C dynamics and the terrestrial feedback to climate warming.

During the past several decades, the regulation of the priming effect by plant roots (plant C input by rhizodeposition), soil (pH and clay content) and microbial properties (microbial biomass, composition and activity) has been studied intensively[8–10]. Despite all the research conducted thus far, our understanding of the dominant priming driver over broad geographic scales is still inadequate since the current studies have been mainly conducted at the site level. Encouragingly, the research community has now begun to address this issue by conducting either regional-scale measurements or global-scale syntheses. Specifically, recent experimental evidences have revealed the dependence of the priming on soil nitrogen content[11] and microbial properties[12] at the regional scale. Likewise, the synthesis of site-level observations has also showed associations of the priming effect with the quantity and quality of plant C input[7,13], soil texture[14] and microbial biomass[7] over broad geographic scales. These studies provide a basic understanding of the potential priming drivers over large scales. However, the variables that have been considered only explained 7.7–22.8% of the variations in the priming intensity[7], with substantial variance left unexplained. The large model residuals highlight the inadequate consideration of the critical drivers in current large-scale studies.

Recently, increasing evidence has revealed the vital role of SOM stabilization mechanisms (i.e. chemical recalcitrance and physico-chemical protection) in determining soil C turnover[15,16]. These mechanisms may also mediate the priming effect through the alteration of microbial C demand[17] and the accessibility of soil C sources[18]. However, the role of these SOM stabilization mechanisms (hereafter called SOM stability) in regulating the priming effect as well as their relative importance compared with other factors (i.e. plant, soil and microbial properties) have not been identified over broad geographic scales[12]. Specifically, previous studies have usually characterized SOM recalcitrance indirectly by using the C:nitrogen ratio[12,14] or the dissolved organic C content[19] and have mainly focused on the dependency of the priming effect on the clay content, or aluminium (Al) and iron (Fe) oxides[12,20]. In fact, the SOM molecular composition (e.g. carbohydrate, lipid and lignin contents) can represent chemical recalcitrance more directly[21], and the physico-chemical protection mediated by calcium ($Ca^{2+}$) and soil aggregates may outweigh the role of Al/Fe oxides in neutral and alkaline soils[22]. Due to the interactive effects of SOM stability, plant, soil and microbial properties on SOM decomposition[16], a comprehensive study with systematic measurements of the priming effect together with these potential drivers over a broad geographic scale is highly needed.

In this study, we quantify the relative importance of plant (vegetation productivity and composition), soil (nutrient content, pH, texture and bulk density) and microbial properties (total biomass, composition, structure, enzyme activities and stoichiometry),

and SOM stability (chemical recalcitrance and physico-chemical protection) in regulating the priming effect based on 30 sites along an ~2200 km grassland transect on the Tibetan Plateau (Supplementary Fig. 1). These sites cover a wide precipitation gradient, from 89 mm in arid climates up to 534 mm in humid climates. Across this precipitation gradient, net primary productivity (NPP) varies between 38 and 488 g m$^{-2}$ yr$^{-1}$. Both soil and microbial properties are also highly variable, with soil organic C (SOC) and microbial biomass C ranging from 1.1 to 118 g kg$^{-1}$ and 23 to 1101 mg C kg$^{-1}$, respectively (Supplementary Data 1). Based on the large-scale sampling along these broad environmental gradients, we determine the priming effect induced by $^{13}C$-labelled glucose during a 65-day laboratory incubation, and then combine multiple analytical approaches (acid hydrolysis, biomarker analysis, a two-pool C decomposition model, aggregate fractionation and mineral analysis) to quantify the role of SOM stability in regulating the priming intensity. We further conduct three types of statistical analysis (i.e. partial correlation, variation partitioning analysis and structural equation modelling) to identify the relative importance of the various factors in shaping the regional patterns of the priming effect. We find that SOM stability characterized by chemical recalcitrance and physico-chemical protection exerts a more important role than other factors over a broad geographic scale.

## Results

**Controls of SOM stability over the priming effect.** Both chemical recalcitrance (i.e. the proportion of the labile and recalcitrant SOM factions) and physico-chemical protection (i.e. the proportion of C protected by aggregates, Fe and Al oxides and exchangeable calcium) were determined to characterize the SOM stability, and then used to explore their effects on priming intensity. Our results showed that the priming intensity increased with the chemical recalcitrance of SOM (Fig. 1). Specifically, both the priming intensity and the relative priming effect (as percentage of basal respiration) were negatively correlated with the proportions of labile pool I (composed of non-cellulosic polysaccharides[23]) (all $p < 0.01$, Fig. 1a; Supplementary Fig. 2a), labile pool II (consisting of cellulose[23]) (all $p < 0.05$, Fig. 1b; Supplementary Fig. 2b) and the labile carbohydrate content (all $p < 0.05$, Fig. 1d; Supplementary Fig. 2d). Conversely, increases in the recalcitrant pool (mainly composed of polymers of lipid and lignin[24]) raised the priming intensity (all $p < 0.01$, Fig. 1c; Supplementary Fig. 2c). Likewise, most of the recalcitrant SOM component (e.g. cutin- and suberin-derived compounds) exerted positive effects on the priming intensity (all $p < 0.01$, Fig. 1e–f; Supplementary Fig. 2e–f), although no correlation was found between the priming intensity and lignin-derived phenols (Supplementary Fig. 3a). These results from acid hydrolysis and biomarker analysis were supported by the negative correlation of priming intensity with the proportion of fast C pool estimated by the two-pool C decomposition model ($p < 0.05$, Supplementary Fig. 3b).

Our results also revealed negative relationships between the priming effect and soil physico-chemical protection. Specifically, priming intensities were low in soils with high molar ratios of Fe/Al oxides or exchangeable Ca to SOC (all $p < 0.01$, Fig. 1g–i). Moreover, the priming intensity was positively correlated with the proportion of C occluded in macroaggregates (>250 μm) ($p < 0.01$, Fig. 1j), but negatively related to the proportion of C occluded in microaggregates (53–250 μm) ($p < 0.01$, Fig. 1k) and clay + silt fractions (<53 μm) ($p < 0.05$, Fig. 1l). Notably, the patterns obtained for the relative priming effect were the same as those corresponding to the priming intensity (Supplementary Fig. 2g–l). Taken together, these results demonstrated that regional variations in priming intensity were jointly controlled by SOM chemical recalcitrance and physico-chemical protection.

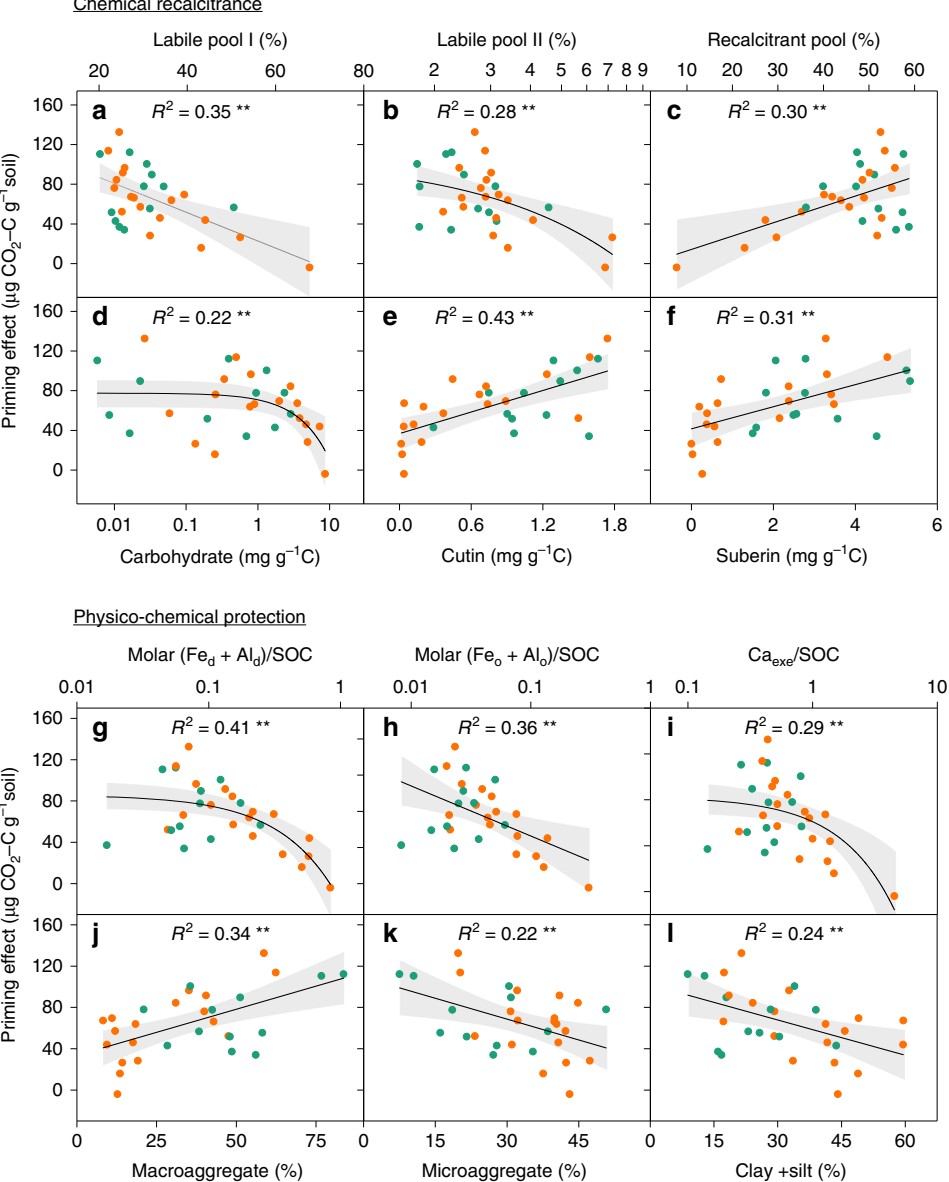

**Fig. 1** Relationships of the priming effect with soil organic matter chemical recalcitrance and physico-chemical protection. Soil organic matter chemical recalcitrance is characterized by the proportion of labile pool I (**a**), labile pool II (**b**), the recalcitrant pool (**c**), relative abundances of carbohydrates (**d**), cutin-derived compounds (**e**) and suberin-derived compounds (**f**). Physico-chemical protection is reflected by the molar ratios of free Fe/Al oxides (**g**) and amorphous Fe/Al oxides (**h**) to SOC, the ratio of exchangeable Ca to SOC (**i**), and the SOC proportions in macroaggregates (**j**), microaggregates (**k**), and clay + silt fractions (**l**). The solid lines represent the fitted ordinary least-squares model, and the grey areas correspond to 95% confidence intervals. A base-10 log scale is used for the x-axis of **b**, **d** and **g**–**i**. The orange dots and green squares represent data in alpine steppe and alpine meadow, respectively. \*\*Significant correlation between the priming effect and the corresponding variables at the 0.01 level

**Importance of SOM stability relative to other properties**. Three statistical methods, including partial correlation analysis, variation partitioning analysis and structural equation modelling, were used to discern the relative importance of SOM stability (i.e. chemical recalcitrance and physico-chemical protection) compared with other factors. Partial correlation analysis showed that, without controlling the role of SOM stability (zero-order in Fig. 2), the priming intensity was closely correlated with plant, soil and microbial properties. However, after controlling SOM stability, the correlation coefficients between the priming intensity and plant, soil and microbial properties decreased by 33.0%, 80.1% and 97.0%, respectively. As a consequence, except for the coverage of sedge and forb, whose correlation with priming retained or increased, most of these factors were no longer

associated with priming effect (Fig. 2). In contrast, the correlation coefficients only decreased by 19.6%, 19.4% and 6.0% between the priming intensity and SOM stability after controlling plant, soil and microbial properties, respectively (Supplementary Fig. 4). In particular, the abundance of cutin-derived compounds was always significantly correlated with the priming intensity, despite its correlation with the priming intensity decreased by 42.9% when the glucose input amount was controlled (Supplementary Fig. 4). Similar results were also found for the relative priming effect (Supplementary Figs. 5–6). These results highlighted that SOM stability was the dominant driver of priming intensity over a continental scale.

Variation partitioning analysis also indicated that SOM stability explained a much greater portion of the variance (38.6%) in

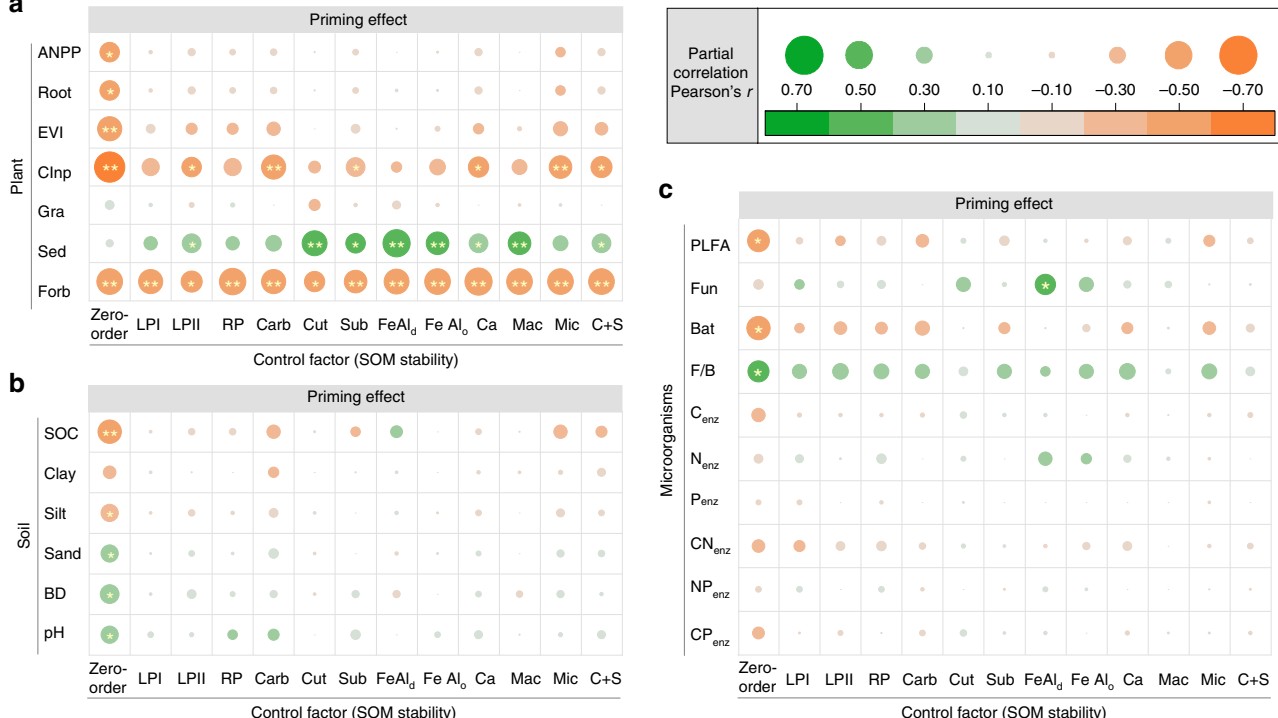

**Fig. 2 Partial correlations between the priming effect and the plant, soil and microbial properties after controlling soil organic matter stability.** The x-axis shows the zero-order (without controlling any factors) and the factors being controlled. The y-axis shows the factors (**a** plant, **b** soil, **c** microbial properties) of which the correlations with the priming effect are examined. The size and colour of the circles indicate the strength and sign of the correlation. Differences in circle size and colour between the zero-order and controlled factors indicate the level of dependency of the correlation between the priming effect and the examined factor on the controlled variable (no change in circle size and colour between the controlled factor and zero-order = no dependency; a decrease/increase in circle size and colour intensity = loss /gain of correlation). ANPP, aboveground net primary productivity; EVI, enhanced vegetation index; CInp, C input amount; Gra, relative coverage of grass; Sed, relative coverage of sedge; Forb, relative coverage of forb; SOC, soil organic carbon content; BD, bulk density; PLFA: total PLFAs; Fun, fungal PLFAs; Bat, bacterial PLFAs; F/B, fungi/bacteria ratio; $C_{enz}$, C-acquiring enzyme activity; $N_{enz}$, N-acquiring enzyme activity; $P_{enz}$, P-acquiring enzyme activity; $CN_{enz}$, C:N ratio of enzyme activity; $NP_{enz}$, N:P ratio of enzyme activity; $CP_{enz}$, C:P ratio of enzyme activity; LPI, labile pool I; LPII, labile pool II; RP, recalcitrant pool; Carb, carbohydrate; Cut, cutin-derived compound; Sub, suberin-derived compound; $FeAl_d$, molar ratio of free Fe/Al oxides to SOC; $FeAl_o$, molar ratio of amorphous Fe/Al oxides to SOC; Ca, ratio of exchangeable Ca to SOC; Mac, proportion of C occluded in macroaggregates; Mic, proportion of C protected by microaggregates; C + S, proportion of C associated with clay + silt fractions. *$p < 0.05$; **$p < 0.01$

priming intensity than plant (6.7%), soil (8.1%) and microbial properties (4.0%) (Fig. 3a). Similarly, this predominant contribution of SOM stability was observed for the relative priming effect (Fig. 3b). Structural equation modelling analysis further revealed that SOM stability and microbial properties were two direct factors that regulated the priming effect. The combination of plant, soil and microbial properties as well as SOM stability explained 54% and 43% of the variance in the priming intensity and relative priming effect, respectively (Fig. 4). Of the factors tested in this study, SOM stability had the largest direct effect (Supplementary Fig. 7). In contrast, plant and soil properties had indirect affects through their associations with SOM stability and microbial properties (Fig. 4). Taken together, these three statistical analyses illustrated that SOM stability was a key factor regulating the priming effect over a broad geographic scale.

## Discussion
Based on multiple analytical and statistical approaches, this study provided empirical evidence that SOM stability, characterized by chemical recalcitrance and physico-chemical protection, had higher predictive power than other factors in explaining regional-scale patterns of the priming effect. The priming intensity increased with the proportion of recalcitrant pool and the content

of recalcitrant components (e.g. cutin and suberin), but decreased with physico-chemical protection by minerals and aggregates (Fig. 5). The regulation of SOM chemical recalcitrance on the priming effect could be attributed to its potential impacts on microbial C and N requirements[25]. On the one hand, compared with soils rich in labile C fractions (e.g. carbohydrates), soils with more recalcitrant polymers usually provide less bioenergetically favourable substrates to microorganisms[18,26], leading to the relatively higher microbial C limitation and a larger proportion of microbial dormancy[25]. Consequently, more microorganisms could be activated by labile C inputs to these soils[25], further stimulating SOM decomposition[4]. On the other hand, high SOM recalcitrance can also trigger great microbial N demand, since the complex molecular structure can hinder the breakdown of N-containing polymers (e.g. chitin) to access available N[27]. The high microbial N requirement could then contribute to $CO_2$ release from these soils by either stimulating SOM decomposition to acquire N (microbial N mining theory)[9] or accelerating microbial N recycling at the expense of C use efficiency[28].

The decrease in the priming intensity with physico-chemical protection by minerals and aggregates could be due to its constraint on SOM availability[15,16]. Specifically, Fe/Al (hydro)oxides have negatively charged functional groups[29] that enable them to participate in ligand exchange reactions and effectively adsorb

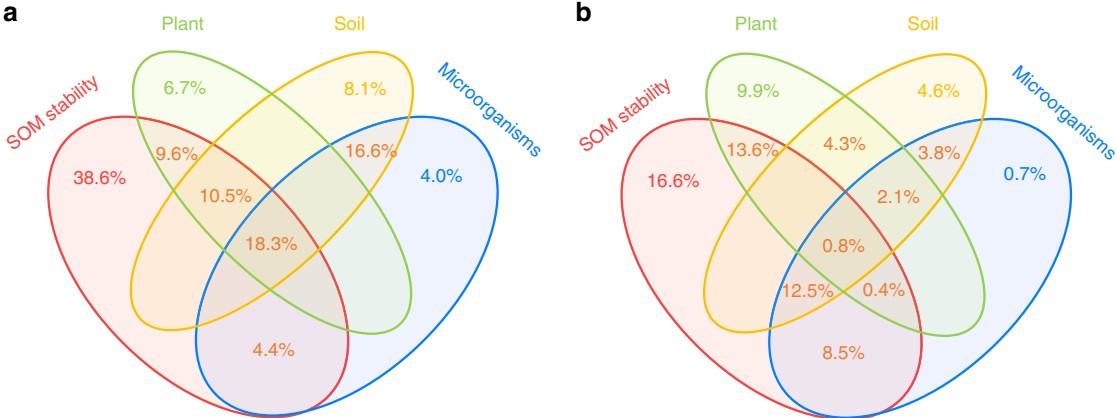

**Fig. 3** Relative contributions of soil organic matter stability, plant, soil and microbial properties to the priming effect. Variation partitioning analysis was conducted to identify the variance in the absolute priming effect (**a**) and relative priming effect (**b**) explained by these four groups of biotic and abiotic factors. SOM, soil organic matter

SOM protecting them against biodegradation[30]. In addition to Fe/Al oxides, $Ca^{2+}$ can also constrain SOM availability by forming both inner- and outer-sphere cation bridging[22], with its protective effect outweighing Fe/Al oxides in neutral to alkaline soils[22]. Additionally, these oxides and cations can accelerate the formation of stable aggregates by their cementation of other minerals[29,30], which results in the compartmentalization of substrates from enzymes and restricts the diffusion of oxygen, thus decreasing SOM decomposition. Consistent with this deduction, the proportion of C associated with clay + silt fractions increased with the molar ratios of metal to SOC across our study area (Supplementary Fig. 8). Taken together, mineral and aggregate protection could inhibit the priming effect through their constraints on organic C availability.

Although both chemical recalcitrance and physico-chemical protection of SOM could regulate the priming effect over a broad geographic scale, the effect of chemical composition depends largely on the degree of physico-chemical protection[16,31–33]. When a relatively larger proportion of SOM is protected by mineral–organic associations and aggregate occlusion, SOM stability could be primarily governed by the degree of protection rather than chemical recalcitrance[32]. Nevertheless, chemical recalcitrance still regulates microbial decomposition when SOM is weakly stabilized by physico-chemical interactions[34]. This phenonmenon can be observed with dissolved organic C[35], free light-fraction SOM[36] and, to a certain extent, organic components with fewer associations with minerals[26,34]. Given that only 15.8% of the soil C pool was protected by minerals across our study area[37], both SOM decomposition and the associated priming effect were jointly affected by the chemical composition and mineral protection.

In addition to SOM stability, plant properties (i.e. plant productivity and community composition) were also important in regulating the priming effect, and exerted strong indirect impacts through their associations with SOM stability (Fig. 4). Particularly, all the variables of SOM stability were significantly correlated with 75% of the plant properties (e.g. aboveground net primary production (ANPP), enhanced vegetation index (EVI), C input rate and the coverage of grass) (Supplementary Fig. 9). The strong dependence of SOM stability on plant properties could be ascribed to the following aspects. On the one hand, as the main SOM source, the rate of plant C input directly determines the amount of plant-derived C, which could be newly stabilized in soil[32] and affects the decomposition of stable SOM pool via the priming effect[3]. Plant C input can also mediate geochemical processes (e.g. soil acidification), which affect soil pH and the

amount of cations[38], thus indirectly regulating the SOM stability. On the other hand, plant community composition controls the chemistry of plant litter and root exudates[39], thereby directly affecting the chemical recalcitrance of SOM. The organic acids (e.g. oxalic acid) from root exudates can also destabilize SOM by liberating C, which are associated with clay minerals, and thus increase C accessibility[18]. Overall, plant properties can regulate the priming effect through their modification on SOM stability.

Although our study demonstrated the important role of chemical recalcitrance and physico-chemical protection in regulating the priming effect over a broad geographic scale, there are still some limitations that need to be addressed in future studies. Particularly, despite being a frequently used approach in priming experiments, a single pulse of glucose addition with an amount equal to the microbial biomass cannot realistically characterize plant C input in terms of quantity, quality[8,40] and input frequency in natural ecosystems. This single addition may also induce changes in microbial biomass[9], community structure[41] and microbial C use efficiency[42]. Furthermore, due to the decreased microbial diversity[43] and enhanced microbial C starvation during the late stage of incubation[44], this single addition may result in a lower priming effect or even negative priming compared with repeated substrate additions[45]. In situ experiments are thus encouraged to better capture realistic plant C input and elucidate the role of plant properties in regulating the priming effect over a large scale. In addition, the limited ecosystem types and climate zones involved in this study may have also induced uncertainties when generalizing patterns and drivers observed in this study. More empirical studies with diverse ecosystem types (forests, shrubs, etc.) and climate zones (tropical, temperate zone, etc.) are thus needed to further advance our understanding on this issue.

In summary, based on systematic measurements of the priming effect together with plant, soil and microbial properties and SOM stability, this study provides empirical evidence that SOM stability, as regulated by plant productivity and community composition, determines the glucose-induced priming effect over a large scale. The stronger predictability of SOM stability than other factors implies that the priming intensity is also governed by SOM stabilization mechanisms, which is similar to what has been observed for soil C pools[46] and their turnover[47]. The higher predictive power of SOM stability also infers potential uncertainties in priming simulations among current models, in which priming intensity is solely assumed to depend on the amount of plant C input[5,6]. Future modelling studies should thus consider both SOM chemical recalcitrance and physico-chemical

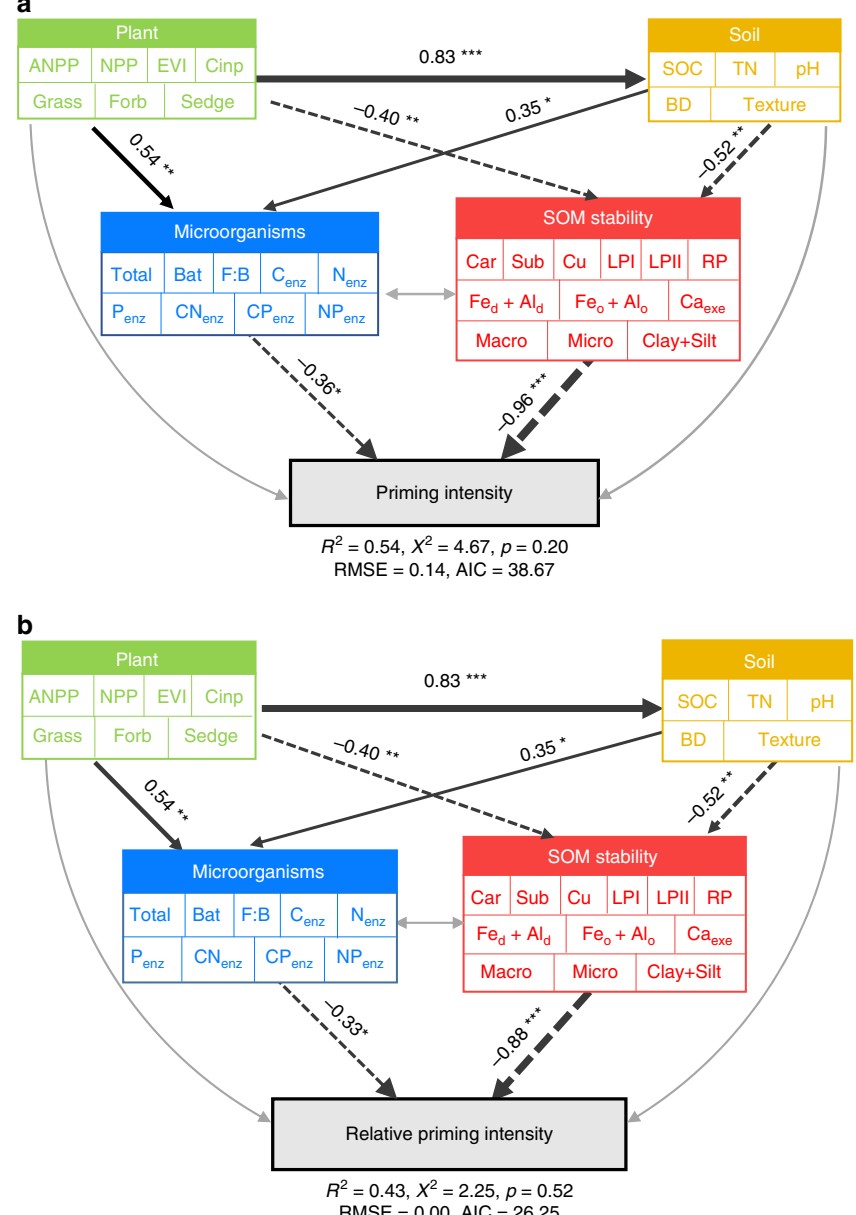

**Fig. 4** Direct and indirect effects of soil organic matter stability, plant, soil and microbial properties on the priming effect. The structural equation modelling was conducted for two forms of priming effect: **a** absolute priming intensity; **b** relative priming intensity. Single-headed arrows indicate the hypothesized direction of causation. Black solid and dotted arrows indicate positive and negative relationships, respectively. Grey arrows indicate insignificant relationship. The arrow width is proportional to the strength of the relationship. Multiple-layer rectangles represent the first component from the PCA conducted for the plant, soil and microbial properties, and soil organic matter stability. The plant properties include aboveground net primary production (ANPP), net primary productivity (NPP), C input amount (Cinp), and the relative coverage of grass, forb and sedge; the soil properties include the soil organic carbon (SOC) content, total nitrogen (TN) content, pH, bulk density (BD) and texture; the microbial properties include the total PLFAs (Total), bacterial PLFAs (Bat), F/B ratio, C-acquiring enzyme activity ($C_{enz}$), N-acquiring enzyme activity ($N_{enz}$), P-acquiring enzyme activity ($P_{enz}$), C:N ratio of enzyme activity ($CN_{enz}$), C:P ratio of enzyme activity ($CP_{enz}$), and N:P ratio of enzyme activity ($NP_{enz}$); the SOM stability includes carbohydrates (Car), suberin-derived compounds (Sub), cutin-derived compounds (Cu), labile pool I (LPI), labile pool II (LPII), the recalcitrant pool (RP), molar ratios of free and amorphous Fe/Al oxides to the SOC ratio ($Fe_d + Al_d$ and $Fe_o + Al_o$), ratio of exchangeable Ca to SOC ($Ca_{exe}$) and the proportions of SOC in macroaggregates (Macro), microaggregates (Micro) and clay + silt fractions. The numbers adjacent to the arrows are the standardized path coefficients

protection to better forecast soil C dynamics under changing environments.

## Methods

**Study area and field sampling**. The Tibetan Plateau is the largest and highest plateau in the world and has broad environmental gradients[48]; as such, this area serves as an ideal platform for exploring the dominant drivers of the priming effect over a broad geographic scale. Specifically, the climate is characterized as cold and dry across the plateau, with a southeast to northwest precipitation gradient ranging from 84 to 593 mm per year. The mean annual temperature in this area varies from −4.9 to 6.9 °C[49]. The dominant ecosystem on the plateau is alpine grassland, which shifts from alpine steppe in the northwestern area to alpine meadow in the southeastern area[48]. Of the two major grassland types, the alpine steppe is mainly dominated by *Stipa purpurea* and *Carex moorcroftii* and is characterized by low precipitation amounts, low plant productivity and a low soil C content. In contrast, the alpine meadow, dominated by *Kobresia pygmaea*, *K. humilis* and *K. tibetica*, receives relatively high amounts of precipitation and corresponds to high plant

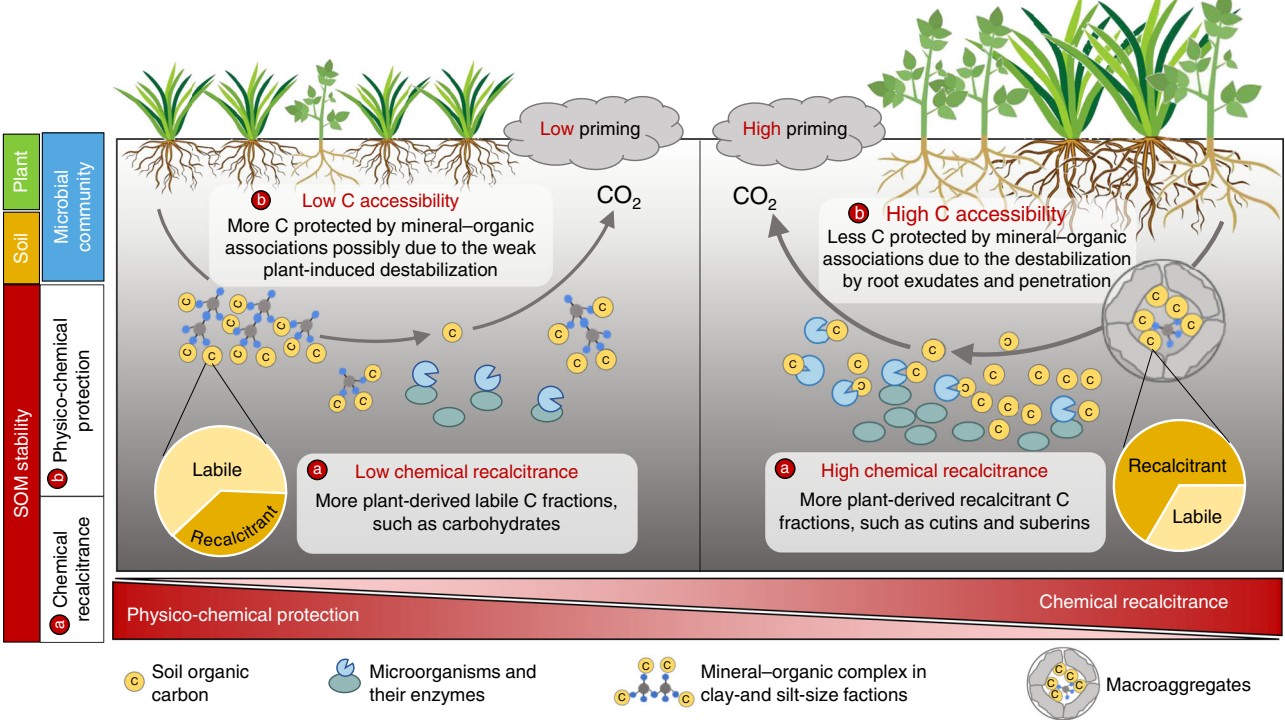

**Fig. 5** Soil organic matter stability, as regulated by plant C input and community composition, determines the priming effect. With the increase of plant productivity and forb cover, more plant-derived recalcitrant SOM fractions (e.g. cutins and suberins) accumulated in the soil. The proportion of C occluded in macroaggregates increased at the expense of C associated with microorganisms and clay + silt fractions, and the physico-chemical protection by aggregations and minerals became weaken. Consequently, either (**a**) the increased chemical recalcitrance or (**b**) the reduced physico-chemical protection (increased C accessibility) could lead to the stronger priming intensity. Notably, this conceptual diagram mainly focused on the most important direct (i.e. soil organic matter stability) and indirect factors (i.e. plant properties) derived from the structural equation modelling analysis

productivity and a high soil C content. The soil orders in this region include Cambisol, Calcisol, Chernozem, and Kastanozem according to the World Reference Base for Soil Resources[50].

To cover these broad environmental gradients, we sampled 30 sites (including 18 steppe sites and 12 meadow sites) along an ~2200 km grassland transect during the growing seasons (between early July and early September) in 2013 and 2014 (Supplementary Fig. 1; Supplementary Table 1). These sites were set throughout the geographical extent of alpine grasslands along the major road at intervals of a certain distance[49], covering a wide range of climates (mean annual temperature: −3.7 to 6.9 °C, mean annual precipitation: 89–534 mm), plant productivity levels (NPP: 38–488 g m⁻² yr⁻¹), soil properties (organic C: 1.1–118 g kg⁻¹; pH: 6.2–9.4; silt content: 5–52%; bulk density: 1.0–1.5 g cm⁻³) and microbial properties (microbial biomass C: 23–1101 mg C kg⁻¹) across the plateau (Supplementary Fig. 10; Supplementary Data 1). Notably, the major considerations for selecting these sampling sites were based on the precipitation gradient, because precipitation induced regional variations in vegetation[48] and soil properties[51] across our study area, which further determined the pattern of soil microbial community[52].

Field sampling was conducted according to a standardized sampling method[49]. Briefly, at each site, we set up five 1 m × 1 m quadrats at the corners and the centre of a 10 m × 10 m plot. For each quadrat, we estimated the mean coverage for each vascular species and then determined the ANPP by clipping the aboveground plants at ground level. Moreover, soil samples in the top 10 cm were collected from three quadrats along a diagonal line in the 10 m × 10 m plot. Considering the high cost of laboratory analyses, as done in previous studies[11,12], three replicates were mixed as one composite sample for subsequent analyses. The bulk density samples were obtained using a standard container with a fixed volume size of 100 cm³ and oven-dried at 105 °C to obtain their masses. The oven-dried soil mass and container volume were then used to quantify the bulk density.

**Glucose addition experiment and uncertainty analysis**. We conducted an isotope-labelled glucose addition experiment including two substrate addition treatments (deionized water or ¹³C-labelled glucose) to determine the priming effect. For each site, fresh soil samples (20 g dry weight) were incubated in triplicate at a fixed soil moisture level (60% of the water holding capacity). Given that the soils varied substantially in terms of their moisture during the field collection, fixing the soil moisture to the same percentage of the water holding capacity facilitated the comparison of microbial processes among multiple sites[53]. Meanwhile, the gravimetric water content during the laboratory incubation was closely

correlated with the corresponding value during the field sampling (Supplementary Fig. 11), reflecting that there was still a spatial gradient in soil moisture even after the adjustment.

To simulate plant C input, a uniformly labelled ¹³C-glucose solution (3.0 atom% ¹³C) was evenly added to the incubated soils after 7 days of pre-incubation. Given the strong dependence of microbial biomass C on plant C input (Supplementary Fig. 12), the amount of glucose addition was set as 100% microbial biomass C at each site. This C input was approximately equal to 22% of NPP within the natural range (10–30% of NPP) of root exudates[54]. Correspondingly, an equivalent volume of deionized water was added as a control. Additional five empty bottles were incubated simultaneously as blanks. Respiration was measured during a 65-day incubation (days 1, 3, 8, 15, 35 and 65) at 15 °C (average temperature of summer season across the study area) after the glucose addition. Before each measurement, the CO₂-free air was flushed for 20 min. Thereafter, all bottles were placed in an incubator for 2 h or 8 h depending on the day of incubation, and 50 ml headspace gas samples were then collected. These gas samples were further used to measure the CO₂ concentration with gas chromatography (Agilent 7890A, Palo Alto, CA, USA), and also determine the δ¹³C abundance with isotopic ratio mass spectrometry (IRMS 20-22, SerCon, Crewe, UK).

To eliminate background δ¹³C noise, we calibrated the C isotope composition of CO₂ respiration (at%$_{respired}$) from both the glucose-treated soils and control soils following Eq. (1):

$$at\%_{respired} = (at\%_{soil} \times C_{Soil} - at\%_{blank} \times C_{blank})/(C_{soil} - C_{blank}),$$ (1)

where at%$_{soil}$ and at%$_{blank}$ are the C isotope compositions (in atom% ¹³C) of CO₂ in the bottles with soil samples and in the blank bottle, respectively; $C_{soil}$ and $C_{blank}$ are the CO₂ concentration (mmol CO₂ mol⁻¹) in the corresponding bottles.

To explore the potential effects of carbonates on CO₂ production and its δ¹³C, we determined the changes in the carbonate content and its δ¹³C value before and after the 65-day incubation and also conducted two carbonate addition experiments (Supplementary Methods). The inorganic C content and its isotope value were determined with a solid TOC (total organic carbon) analyser (multi EA 4000, Analytik Jena, Germany) and an isotopic ratio mass spectrometry (Thermo Scientific 253 Plus, USA), respectively. Our analyses showed that there were no significant changes in either carbonate content or the corresponding δ¹³C value before and after the 65-day incubation (Supplementary Fig. 13). Moreover, CaCO₃ addition did not significantly affect the cumulative CO₂ release during the 30-day incubation in neither carbonate addition experiment (Supplementary Fig. 14).

Particularly, the $^{13}$C-labelled $CaCO_3$ experiment revealed that $CaCO_3$-derived $CO_2$ only accounted for 1.0% of the total $CO_2$ release. These analyses demonstrated a minor contribution of carbonates to the $CO_2$ production and also its $\delta^{13}$C. Therefore, as done in most priming studies on soil with pH > 7.0[11,55], the effects of soil carbonates on $CO_2$ release were considered limited in this study. The fractions of $CO_2$-C derived from the added $^{13}$C-glucose ($f_{glucose}$) and from the SOM pool ($f_{SOM}$) were then determined by Eqs. (2) and (3).

$$f_{glucose} = (at\%_{treat} - at\%_{control})/(at\%_{glucose} - at\%_{SOM}), \quad (2)$$

$$f_{glucose} + f_{SOM} = 1, \quad (3)$$

where $at\%_{treat}$, $at\%_{control}$, $at\%_{glucose}$ and $at\%_{SOM}$ are the C isotope compositions (in atom% $^{13}$C) of $CO_2$ from the glucose-treated soil, control soil after correction by Eq. (1), added glucose and SOM, respectively.

The priming effect ($\mu$g $CO_2$-C g dw$^{-1}$) and relative priming effect (%) were further determined following Eqs. (4) and (5).

$$Priming\ effect = C_{treat} \times f_{SOM} - C_{control}, \quad (4)$$

$$Relative\ priming\ effect = (C_{treat} \times f_{SOM} - C_{control})/C_{control}, \quad (5)$$

where $C_{treat}$ and $C_{control}$ are the total $CO_2$-C ($\mu$g $CO_2$-C g$^{-1}$ soil) from the glucose-treated soil and control soil, respectively.

To quantify uncertainties in the priming effect, we used the Monte Carlo approach to estimate its 95% confidence interval[13,56]. Specifically, we first quantified the uncertainties in the SOM-derived $CO_2$ release for each time's $CO_2$ and $\delta^{13}$C measurements under both the glucose treatment and control conditions. The mean values and standard deviations (SDs) of $CO_2$ release under the control conditions were calculated based on triplicate samples. In contrast, the SD of SOM-derived $CO_2$ release under the glucose treatment was calculated according to the following error propagation characterized in Eq. (6)[44], in which the uncertainties in both the $CO_2$ flux and $\delta^{13}$C measurements were considered simultaneously:

$$\sigma_{SOM} = \sqrt{\left(\frac{\sigma_{C_{treat}}}{C_{treat}}\right)^2 + \left(\frac{\sigma_{f_{SOM}}}{f_{SOM}}\right)^2}, \quad (6)$$

where $\sigma_{SOM}$ is the SD of the SOM-derived $CO_2$ flux in the glucose treatment, $\sigma_{C_{treat}}$ and $\sigma_{f_{SOM}}$ are the SDs of the total $CO_2$ fluxes in the glucose-treated soil and the SOM pool fraction, respectively, and $C_{treat}$ and $f_{SOM}$ are their respective mean values.

After obtaining the means and SDs of SOM-derived $CO_2$ release under both the glucose treatment and control conditions, we performed 1000 Monte Carlo simulations for the priming effect during each time's measurement. First, the SOM-derived $CO_2$ releases in both the glucose treatment and control conditions were randomly generated based on normal distributions using the above-mentioned means and SDs. The priming effect was then calculated as the difference in the SOM-derived $CO_2$ release between the glucose treatment and the control. The cumulative priming effect was further estimated by integrating the absolute priming effect over the incubation time. Finally, we calculated the 95% confidence interval of the cumulative priming effect based on a total of 1000 estimates derived for each study site.

**Measurements of biotic and abiotic variables.** Given plant, soil and microbial properties as well as SOM stability being the major drivers of SOM decomposition and the priming effect[16], to facilitate our interpretations, we classified all factors into four groups to explore their roles in regulating the priming effect: plant properties (NPP, ANPP, root biomass, EVI, C input amount and relative coverage of grasses, forbs and sedges), soil properties (SOC content, total N content, pH, clay, silt, sand, bulk density and soil order), microbial properties (bacterial phospholipid fatty acids (PLFAs), fungi PLFAs, the ratio of fungi to bacteria (F/B) ratio, C-, N-, P-acquiring enzyme activities, C:N ratio of enzyme activity, N:P ratio of enzyme activity and C:P ratio of enzyme activity) and SOM stability (labile C pool I, labile C pool II, recalcitrant pool, carbohydrate, lignin-derived phenol, cutin-derived compounds, suberin-derived compounds, the C associated with macro-aggregates, microaggregates and the clay + silt fraction, molar ($Fe_d + Al_d$)/SOC, molar ($Fe_o + Al_o$)/SOC and $Ca_{exe}$/SOC).

**Plant, soil and microbial properties.** Of the plant properties, the relative cover of the three major functional groups (grass, forb and sedge) was derived from our own community investigation. Both the EVI (2010–2014) and NPP (2000–2014) estimates were obtained from Moderate Resolution Imaging Spectroradiometer data (http://modis.gsfc.nasa.gov). The root biomass was estimated from the aboveground biomass using the isometric relationship observed across the Tibetan alpine grasslands[57]. Additionally, the amount of glucose-C addition was set to 100% of the microbial biomass C for each site, which was analysed using the chloroform fumigation method with a conversion factor of 0.45.

For soil properties, we determined a series of soil parameters reflecting the basic soil environmental conditions. The SOC content was determined by the potassium dichromate oxidation method[58]. The total nitrogen content was analysed using an elemental analyser (Vario EL III, Elementar, Germany) after air drying. The soil pH

was measured by a pH electrode (PB-10, Sartorius, Germany) in a 1:2.5 soil:water suspension. The soil texture was determined using a particle size analyser (Malvern Masterizer 2000, UK) after removing the organic matter and carbonates by hydrogen peroxide and hydrochloric acid, respectively. The water holding capacity was determining by rewetting the soil samples to saturation and then oven-drying them at 105 °C for 24 h.

Although the amount of glucose addition equalling to 100% of the microbial biomass had been widely adopted in priming experiments[59], the method may alter the potential role of microbial properties in regulating the priming effect. To minimize this potential impact, we used per unit soil weight and % of flux as the unit of the priming effect instead of using per unit C added to avoid the removal of the C input effect. Moreover, we used the microbial community structure (i.e. F/B), microbial enzyme activity (i.e. $\beta$-1,4-glucosidase, BG; $\beta$-1,4-$N$-acetylglucosaminidase, NAG; leucine aminopeptidase, LAP; and acid/alkaline phosphatase, AP) and their stoichiometric ratios as additional microbial properties. Specifically, PLFAs were extracted from the soil following the protocol provided by Bossio and Scow[60], and used to evaluate the abundance of bacteria and fungi as well as the F/B ratio. The activities of the C-acquiring (BG), N-acquiring (NAG and LAP) and P-acquiring enzyme (AP) were assayed using fluorometric techniques by constructing calibration curves for each sample[61]. Based on these measurements, we calculated the ratios of ln(BG):ln(NAG + LAP), ln(NAG + LAP):ln(AP) and ln (BG):ln(AP) activities to represent the relative microbial nutrient demand. The detailed methods for the PLFA and enzyme activity analyses are provided in the Supplementary Methods.

**Identification of SOM stability.** SOM stability was quantified by two types of stabilization mechanisms: SOM chemical recalcitrance and physico-chemical protection. For SOM chemical recalcitrance, we used the relative abundances of the labile and recalcitrant SOM pools measured by the acid hydrolysis[23], the relative abundances of particular SOM components determined by biomarker analysis[21] and the proportion of fast C pool inversed from the two-pool C decomposition model[62]. Specifically, the proportions of labile pool I, labile pool II and the fast C pool, and the content of carbohydrates were used to represent the labile SOM fractions, while the proportion of recalcitrant C pool and the relative abundances of lignin-, cutin- and suberin-derived compounds were used to reflect the recalcitrant SOM fractions. For physico-chemical protection, the proportion of C in macro-aggregates, microaggregates and clay + silt fractions was determined by soil C fractionation[46], and the ratios of metal to SOC (i.e. free and amorphous Fe and Al oxides: molar $Fe_d + Al_d$/SOC, molar $Fe_o + Al_o$/SOC; and exchangeable calcium: $Ca_{exe}$/SOC) were applied[63].

**Acid hydrolysis method.** The soil labile C pool (composed of polysaccharides and cellulose) and the recalcitrant C pool (composed of wax-derived long-chain ali-phatics and aromatic components) were determined by the acid hydrolysis method[24]. Briefly, soil samples were first hydrolysed with 2.5 M $H_2SO_4$ at 105 °C for 30 min. The hydrolysates were centrifuged and decanted. The residue was washed with distilled water and the supernatant was added to the hydrolysate, which was regarded as labile pool I. The remaining residue was further hydrolysed with 13 M $H_2SO_4$ and shaken overnight at room temperature. Subsequently, the distilled water was added to dilute the acid concentration to 1 M, and the sample was hydrolysed at 105 °C for 3 h. The second hydrolysate was regarded as labile pool II. The remaining soil residue was rinsed twice with distilled water and dried at 60 °C. This fraction was considered the recalcitrant SOM pool. The relative abundances of labile pool I and II and the recalcitrant pool were calculated as the ratio of SOC in each pool to the total SOC[24].

**SOM biomarker analysis.** The biomarker extraction and analysis followed the standard procedures, in which sequential chemical extractions (solvent extraction, base hydrolysis and CuO oxidation) were conducted to separate the solvent-extractable compounds, cutin- and suberin-derived compounds and lignin-derived phenols, respectively[21]. Of the solvent-extractable compounds, we focused on the most labile C fractions (carbohydrates). The detailed chemical extraction and analysis are provided in the Supplementary Methods.

**Soil C decomposition model.** The proportion of fast and slow C pool, another proxy for SOM chemical recalcitrance, were estimated by a two-pool model[21,62]. Specifically, we applied the two-pool model to each of the 30 sites in the control group as follows:

$$R(t) = k_1 f_1 C_{tot} e^{-k_1 t} + k_2 (1 - f_1) C_{tot} e^{-k_2 t}, \quad (7)$$

where $R(t)$ is the $CO_2$ emission rate (mg C g$^{-1}$ SOC day$^{-1}$) at time $t$, $C_{tot}$ is the maximum C loss percentage (i.e. 100% = 1000 mg C g$^{-1}$ SOC), $f_1$ is the proportion of the fast pool, and $k_1$ and $k_2$ are the decay rates of the fast and slow pools (day$^{-1}$), respectively. The three parameters ($f_1$, $k_1$ and $k_2$) were estimated by the Markov chain Monte Carlo (MCMC) approach[62]. Before applying the MCMC approach, the prior parameter range in the initial model (Supplementary Table 2) was set as wide as possible to cover the possibilities for all study sites. After the MCMC simulations, maximum likelihood estimates were used to quantify the well-constrained parameters[21], while mean values were calculated for the poorly

constrained parameters (Supplementary Table 3). The data-model comparison revealed good model performance for the 30 study sites ($R^2 = 0.95$; RMSE = 0.22, Supplementary Fig. 15). Notably, a two-pool model rather than a three-pool model was chosen because the relatively short-term (65 days) incubation made it less possible to estimate the true turnover time of the passive C pool in the three-pool model.

**Physical protection parameters.** The proportion of C in the different functional SOM fractions was used to characterize the role of physical protection by aggregates. The functional C fractions were obtained following the method based on the conceptual C fraction model[46]. Briefly, soil C was fractionated into macroaggregate-associated C (>250 μm), microaggregate-associated C (250–53 μm) and non-aggregated silt + clay (<53 μm)[46]. The isolated fractions were analysed for their SOC contents using an elemental analyser (muti EA 4000) after removing the inorganic C with 1 M HCl. Based on these C fractions and the SOC contents in the bulk soil, we calculated the proportion of C associated with the different fractions.

**Mineral protection parameters.** The contents of Fe and Al oxides and exchangeable Ca were used to calculate the mineral protection parameters. The reactive Fe and Al contents were determined by the citrate-bicarbonate-dithionite method ($Fe_d + Al_d$) and acid oxalate extraction ($Fe_o + Al_o$)[18,64]. The $Fe_d + Al_d$ represents the amount of pedogenic Fe and Al within oxides, silicates and organic complexes, whereas $Fe_o + Al_o$ originates from poorly crystalline mineral complexes[18,64]. The exchangeable Ca was extracted by $NH_4Cl$-ethanol[65]. Based on these measurements and the SOC content in bulk soil, the molar ratios of metals to SOC (molar ($Fe_d + Al_d$)/SOC and molar ($Fe_o + Al_o$)/SOC) and the ratio of exchangeable Ca to SOC ($Ca_{exe}$/SOC) were calculated to represent the potential of minerals to adsorb SOC[63].

**Statistical analyses.** We first conducted ordinary least squares regression to evaluate the relationships between the priming effect and SOM stability. We then used three types of statistical models (partial correlation, variation partitioning analysis and structural equation modelling) to test whether the importance of SOM stability was maintained when accounting for other three types of factors.

Given the strong connections and inter-correlations among the various factors (Supplementary Fig. 9), partial correlation was first conducted to evaluate the relationships between the priming effect and the various factors[46]. For example, after controlling every single variable of SOM stability, we examined the relationships of the priming effect with plant, soil and microbial properties. Similarly, plant, soil and microbial properties were separately set as the controlling factors to explore the relationship between the priming effect and SOM stability. The greater the difference in the partial correlation coefficient between the zero-order and controlling correlation, the stronger the effect of the factor being controlled[46]. Moreover, the relatively greater difference also indicated the stronger interaction effects among these predictors. These analyses were conducted using the packages ggm and psych of the R statistical software v.3.2.4 (R Development Core Team, 2016).

Variation partitioning analysis that partitioned the variance shared by all factors[66] was then used to quantify the unique contribution of each group of factors. A negative value in the variance explained for a group of factors was interpreted as zero, which indicated that the explanatory variables explained less variation than random normal variables[66,67]. The variation partitioning analyses were conducted with the R package vegan v.3.2.4 (R Development Core Team, 2016).

Structural equation modelling was further used to evaluate the direct and indirect relationships between the priming effect and SOM stability, plant, soil and microbial properties. This approach can partition the direct and indirect effects that one variable may have on another[68,69] and is therefore useful for exploring complex relationships in natural ecosystems. Owing to strong correlations among the factors within each group, we conducted principal component (PC) analysis to create a multivariate functional index before structural equation modelling construction. The first component (PC1), which explained 49.7–66.0% of the total variance for these four groups, was then introduced as a new variable to represent the combined group properties into the subsequent analysis (Supplementary Table 4). The fit of the final model was evaluated using the model $\chi^2$ test and the root mean-squared error of approximation. The structural equation modelling analyses were conducted using AMOS 21.0 (Amos Development Corporation, Chicago, IL, USA).

## Data availability

All plant, soil and microbial predictors used in this study are available as a supplementary file (Supplementary Data 1). Additional data that support the findings of this study are available from Y.Y. upon reasonable request.

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

## Acknowledgements
We are grateful to the members of the IBCAS sampling campaign teams for their contributions in the field investigation. We also appreciate Dr. Yinghui Wang and Mr. Xinyu Zhang at Peking University for their assistance in the biomarker analysis, Dr. Junyi Liang at the University of Oklahoma for providing the two-pool carbon decomposition model, Prof. Xingliang Xu at the Institute of Geographic Sciences and Natural Resources Research, Chinese Academy of Sciences (CAS) and Dr. Yunfeng Peng at the Institute of Botany, CAS for providing helpful comments on an early version of this manuscript. This work was supported by the National Key R&D Program of China (2017YFC0503903), the Second Tibetan Plateau Scientific Expedition and Research (STEP) program (2019QZKK0302), National Natural Science Foundation of China (31770557, 31825006, 41877046 and 31922054), Youth Innovation Promotion Association of the Chinese Academy of Science (2017109) and Key Research Program of Frontier Sciences and Chinese Academy of Sciences (QYZDB-SSW-SMC049). Y.K. was supported by the "RUDN University program 5-100" and Russian Science Foundation project 19-77-30012.

## Author contributions
Y.Y., L.C., B.Z. and L.L., designed the research. L.L., L.C., S.Q., G.Y., K.F., P.C. and Y.X. performed the experiments. L.C. performed the C decomposition simulations with the two-pool C decomposition model; L.C. analysed the data. L.C., Y.Y., S.Q., Y.K., B.Z. and L.L. wrote the manuscript with input from the other authors.

## Competing interests
The authors declare no competing interests.
