## [Peer Review File · Nature Communications]

Reviewers' comments:

Reviewer #1 (Remarks to the Author):

General comments:

The study by Chen et al investigates whether plant, microbial or soil organic matter (SOM) variables control the magnitude of priming effects in soils sampled from the Tibetan Plateau. The results demonstrate that SOM decomposability and availability are better predictors of the priming effects than other variables that have been more commonly used in previous studies. I think this is an important finding that is likely to advance the understanding of soil organic matter priming.

I did however, find the reference to 'traditional' plant, soil and microbial predictors a little unhelpful as it implies that the soil community was ignoring soil organic matter quality and availability in the study of priming effects and I do not consider this to be the case (for example, papers cited in this study have been longing at this issue for more than a decade). Rather, I think the new results provide a well-presented analysis of the relative importance of different types of variables in Tibetan Plateau ecosystems, and presents a compelling case for the importance of SOM quality and availability versus plant and microbial factors. It therefore adds important understanding of the relative roles of different potential control factors, helping to test competing hypotheses and it is not necessary to imply SOM quality and availability have been largely ignored to date.

It is always challenging in an analysis of a large number of highly inter-correlated variables to demonstrate which the key variables are, and to demonstrate cause and effect. The analysis could obviously change if new variables were added (more information on the quality of plant inputs, for example), but I found the analysis presented in Figure 2 to be very effective and convincing. However, I would like to request a couple of further analyses be carried out to demonstrate that the results are robust:

1. Firstly, the priming effects are presented per unit soil weight. Priming effects can also be presented as a percentage stimulation of the flux, per unit SOM, or even per unit C added. It would be helpful to demonstrate that the conclusions hold irrespective of the units the priming effects are presented in and are not simply related to SOM quality and availability because they are presented per unit soil mass.
2. In terms of the nutrient mining hypothesis, I was surprised that potential enzyme activity ratios did not seem to be included as microbial predictors. The ratios of C:N, C:P and N:P cycle

enzymes may provide information on the extent of microbial C versus nutrient limitation and this would seem more likely to be informative than the raw potential activity measurements.

3. The rate of addition was made proportional to initial microbial biomass. I recognise that some kind of standardisation is necessary and using microbial biomass makes sense. However, by standardising per unit microbial biomass this may already reduce the potential importance of microbial variables in controlling the magnitude of priming effects. This should be discussed in the paper.

In summary, I found this to be an interesting and potentially important study but would suggest that the authors need to carry out a few more analyses to demonstrate the robustness of the conclusions and would advise against using the term 'traditional' and suggesting that only plant and microbial variables have been investigated to date. Rather, the study would be better framed as a detailed evaluation of the relative importance of plant, microbial and SOM variables in controlling the magnitude of priming effects.

Specific comments:

Lines 277-315: How frequently were the flux and isotopic measurements made and did the magnitude or sign of the priming effects change with time during the 65 day incubation?

Lines 326-327: Are there no data on plant species composition. E.g. relative coverage of grasses, sedges, shrubs etc?

Lines 342-347: The ratios of potential enzyme activities associated with carbon, nitrogen and phosphorus cycles may provide information on how carbon or nutrient limited the microbes were. In terms of testing hypotheses related to nutrient mining, these ratios may be more valuable than individual enzyme activities.

Reviewer #2 (Remarks to the Author):

Chen et al. evaluated soil priming across a gradient of sites in the Tibetan Plateau. I acknowledge the amount of work that went into this project. However, I had a difficult time with the manuscript for the reasons given below.

To start with, I don't completely accept the justification given several times, that "none of previous studies has explicitly addressed the relationship between soil C decomposability and priming effect over the broad geographic scale" (lines 74-76). There are numerous related gradient studies (e.g., Madagascar, <https://www.nature.com/articles/ismej2017178>). Why are these studies not relevant?

More importantly, the paper doesn't do a good job of explaining the 2200-km gradient, either its characteristics or how it relates to the study system. The gradient is described in this way: "The sampling sites covered a wide range of climate, plant productivity and soil conditions affecting soil C stock and its decomposability." Why and how were these 90 sites chosen? How do soil properties change along this gradient, in particular texture and other physical properties. Why, for instance, would you necessarily incubate all soils with fixed soil moisture (60% water holding capacity) (line 280) if there are textural, and hence water potential, differences? It feels at times that the authors were designing an experiment to reach a particular conclusion.

The words "error" and "statistics" are barely mentioned in the manuscript and not at all in the supplement. Much more work and description is needed here for the paper to be acceptable.

The writing will need considerable work for grammar and clarity before publication, regardless of the outcome of review. That statement is not a reflection on the review or a factor to weigh for acceptance or rejection.

I don't find Figure 2 to be very helpful. It is difficult to read and somewhat difficult to interpret.

Some relevant papers:

Meta-analysis: <https://www.sciencedirect.com/science/article/pii/S0038071716301560>

Priming in a similar steppe in China: <https://onlinelibrary.wiley.com/doi/full/10.1111/oik.01728>

Additional comments:

It's my understanding that the authors split all predictors into four groups: plant factors, soil factors, microbial factors, and soil C decomposability. It seems that a few measurements are missing from the first three groups, and the uniqueness of soil C decomposability as a predictor is unclear.

For the 'plant factor', we know belowground inputs (e.g., root biomass and exudation) play a key role in SOM formation and in situ priming. While the transect was dominated by similar vegetation (namely, steppe and meadow grasslands), would you expect significant differences in below-ground biomass across sites? For example, different root biomass and/or rooting depths, depending on climate. Do grass species change across the transect?

The 'soil factor' does not include any information on soil mineralogy or other inherent soil characteristics. I would recommend including soil texture (% clay and silt) and soil order, at the very least, as a part of this factor. The authors could also include bulk density.

With 'soil C decomposability' as a combination of chemical quality (e.g., lignin, etc.) and mineral protection, it seems that it would in fact be driven by underlying soil, plant, and microbial factors. The uniqueness and usefulness of considering this as a novel 'non-traditional factor' for analyzing priming is unclear to me.

Other questions/comments:

- What were the decay rates obtained for the two-pool model? Was a unique solution obtained, or what was the distribution? Was there an uncertainty in the size of the fast and slow pools? How did this, and other measurement uncertainties, follow through the later statistical analyses?

- Do the priming effect measurements (e.g., Fig. 1) have error associated with them? How much did the incubation results (performed in triplicate) vary?

- How did you decide to add the ^{13}C glucose equivalent of 100% of the microbial biomass at each site, as opposed to a different proportion or an addition weighted on the relative amounts of SOC or plant inputs across the sites?

- Do you expect that a large addition of glucose, a low molecular weight 'sweet', could shift the microbial community? Does this limit the generalizability of the results?

- It seems the glucose addition was performed as a pulse at the beginning of the incubation. Can you comment on the potential role and implications of C starvation during the course of the 65-day experiment?

- The term 'mineral-affected' is a bit confusing, and I have not seen it used in the literature. I understand the intent — that the micro-aggregate size fraction is influenced by the formation of underlying mineral-associations and sticky microbial compounds, and therefore they are both in some way 'mineral-affected'. However, these two fractions are very different mechanistically — namely, through physical vs. chemical protection. I would suggest breaking up this pool into clay+silt vs. micro-aggregate contributions, or I would use a different term.

- How would this information be used to inform models? While priming is recognized to be an important feedback and efforts are being made to capture this response in models, it is still debated how to mechanistically or implicitly incorporate this feedback in models. What do the results tell us about mechanisms and correct representations? How can these reported correlations be used?

- Furthermore, such SOM models are often applied across soil and vegetation types. Do you expect your results to be generalizable to other vegetation types, e.g., forests?

Reviewer #3 (Remarks to the Author):

The manuscript by Chen et al is an interesting attempt to estimate the importance of different carbon stabilization process (chemical recalcitrance and mineral interactions) on priming intensity. Priming is considered as an important process controlling soil carbon fluxes but we still lack the understanding of the mechanisms behind. Consequently, predicting the priming intensity is very challenging.

I appreciate the efforts made by the authors to sample over a large region (the Tibetan plateau) but I have several concerns mainly about methodological aspects that should be considered before any publications.

1. A large part of the manuscript relate priming and chemical recalcitrance but at any moment, the important debate occurring in the soil science about chemical recalcitrance (Dungait et al., 2012; Kleber, 2010; Marschner et al., 2008; Mikutta et al., 2006) is mentioned. Even though, the concept is not totally abandoned by the soil science community it is largely criticized (Lehmann and Kleber, 2015; Schmidt et al., 2011) and more and more evidences tends to suggest that this concept is misleading (e.g. Klotzbücher et al., 2011). The authors may disagree and think the concept is still useful but they must explain why and mention how their arguments take part of the current debate.

2. To measure priming, the authors scaled the amount of glucose on the microbial biomass but they do all their analyses with priming given in gC-CO₂ g⁻¹ of soil without considering that priming intensity is related to the labile material amended (e.g. Xiao et al., 2015). I suggest to redo the analysis estimating priming per g of C soil and g C-glucose added to be consistent

3. The modelling approach is difficult to understand and it took me several reading to finally get something but I am even not sure that I understood completely. Basically, the authors considered that with such short term incubations it is not possible to estimate the most passive pool and they therefore fixed the C_{tot} parameters of eq. 5 to be 1000mg g⁻¹. Based on this assumptions they estimate the f₁ parameter and the two decay rates of the corresponding pools. C_{tot} is therefore not the total soil C stocks but the sum of the fast and slow pools. Therefore, the fig. S2b is misleading because it suggests that you correlated the priming with the fast pool size whereas it is actually just the fast pool over the sum of the fast and slow pools. Moreover much more information are needed in particular the prior values, the estimated parameters and their confidence interval.

Minor comments:

L49: Don't forget lateral transfer through erosion for instance

L275: How do you stored the samples before incubations?

L285: How do you estimate the microbial biomass? The amount of glucose added as well as the microbial biomass should be given in sup mat

L286: I don't get why you had blank if you started with CO₂-free atmosphere. The incubation flasks were not closed?

L320: I am ok with this assumption because your sampling points are mainly on grassland and steppe which, I guess, are largely composed of annual plants but it should be a bit more explained in the text.

L342: Please rephrase, I guess your talking about the Bosso et al protocol but with the "Nat. Com." style the reading is not straightforward.

Cited references:

Dungait, J. a. J. J., Hopkins, D. W., Gregory, A. S. and Whitmore, A. P.: Soil organic matter turnover is governed by accessibility not recalcitrance, *Glob. Chang. Biol.*, 18(6), 1781–1796, doi:10.1111/j.1365-2486.2012.02665.x, 2012.

Kleber, M.: What is recalcitrant soil organic matter?, *Environ. Chem.*, 7(4), 320, doi:10.1071/EN10006, 2010.

Klotzbücher, T., Kaiser, K., Guggenberger, G. and Kalbitz, K.: A new model for the fate of lignin in decomposing, *Ecology*, 95(5), 1052–1062, doi:10.2307/41151233, 2011.

Lehmann, J. and Kleber, M.: Perspective The contentious nature of soil organic matter, , 0–8, doi:10.1038/nature16069, 2015.

Marschner, B., Brodowski, S., Dreves, A., Gleixner, G., Gude, A., Grootes, P. M., Hamer, U., Heim, A., Jandl, G., Ji, R., Kaiser, K., Kalbitz, K., Kramer, C., Leinweber, P., Rethemeyer, J., Schäffer, A., Schmidt, M. W. I., Schwark, L. and Wiesenberger, G. L. B.: How relevant is recalcitrance for the stabilization of organic matter in soils?, *J. Plant Nutr. Soil Sci.*, 171(1), 91–110, doi:10.1002/jpln.200700049, 2008.

Mikutta, R., Kleber, M., Torn, M. S. and Jahn, R.: Stabilization of soil organic matter: association with minerals or chemical recalcitrance?, *Biogeochemistry*, 77(1), 25–56 [online] Available from: <http://www.springerlink.com/index/C288615565707287.pdf>, 2006.

Schmidt, M. W. I., Torn, M. S., Abiven, S., Dittmar, T., Guggenberger, G., Janssens, I. a., Kleber, M., Kögel-Knabner, I.,

Lehmann, J., Manning, D. a. C., Nannipieri, P., Rasse, D. P., Weiner, S. and Trumbore, S. E.: Persistence of soil organic matter as an ecosystem property, *Nature*, 478(7367), 49–56, doi:10.1038/nature10386, 2011.

Xiao, C., Guenet, B., Zhou, Y., Su, J. and Janssens, I. a.: Priming of soil organic matter decomposition scales linearly with microbial biomass response to litter input in steppe vegetation, *Oikos*, (124), 649–657, doi:10.1111/oik.01728, 2015.

Response to Reviewer #1:

[Comment 1] General comments: The study by Chen et al investigates whether plant, microbial or soil organic matter (SOM) variables control the magnitude of priming effects in soils sampled from the Tibetan Plateau. The results demonstrate that SOM decomposability and availability are better predictors of the priming effects than other variables that have been more commonly used in previous studies. I think this is an important finding that is likely to advance the understanding of soil organic matter priming.

[Response] Thanks for the reviewer's positive comment.

[Comment 2] I did however, find the reference to 'traditional' plant, soil and microbial predictors a little unhelpful as it implies that the soil community was ignoring soil organic matter quality and availability in the study of priming effects and I do not consider this to be the case (for example, papers cited in this study have been longing at this issue for more than a decade). Rather, I think the new results provide a well-presented analysis of the relative importance of different types of variables in Tibetan Plateau ecosystems, and presents a compelling case for the importance of SOM quality and availability versus plant and microbial factors. It therefore adds important understanding of the relative roles of different potential control factors, helping to test competing hypotheses and it is not necessary to imply SOM quality and availability have been largely ignored to date.

[Response] Very good comment! Following the reviewer's comment, **we have deleted the term 'traditional' in the revised MS, and focused more on the relative importance of different factors in driving the priming effect over broad geographic scale as follows** (Page 4-5, line 86-90; line 97-100). We would also like to mention that, **compared to previous studies, this work included more variables to directly characterize the chemical recalcitrance and other physico-chemical protections** (Table R1). Specifically, previous studies usually characterized SOM quality indirectly by using C:N ratio (Luo *et al.*, 2016, Qiao *et al.*, 2016, Razanamalala *et al.*, 2017), DOC content (Chowdhury *et al.*, 2014) and SOC content

(Table R1). Consequently, it remains unknown whether and how different SOM compositions (e.g., carbohydrate, lipid, and lignin) regulate the priming effect over the large scale. Additionally, although recent studies reported the dependency of priming effect on clay content (Huo *et al.*, 2017; Luo *et al.* 2018), aluminium (Al) and iron (Fe) oxides (Razanamalala *et al.*, 2017, Finley *et al.*, 2018), it remains unclear about the relationships between priming effect and calcium (Ca) and aggregate-mediated protection over broad geographic scale. Based on these understandings, we have clearly clarified the novelty in the *Introduction* section of the revised MS (Page 4-5, line 86-97).

Table R1. Comparison of variables used to characterize SOM quality and physico-chemical protection in previous studies and current study.

Factor	Variables used in previous studies	New variables used in this study
SOM quality	C:N ratio (Razanamalala et al. , 2017; Luo et al. , 2018; Perveen et al. , 2019), DOC (Chowdhury et al. , 2014), SOC (Huo et al. , 2017)	Relative abundance of carbohydrate, cutin, suberin, and lignin derived from biomarker analysis; the proportion of labile C pool and recalcitrant C pool derived from acid hydrolysis; the proportion of fast and slow C pool estimated from two-pool model
Physico-chemical protection	clay content (Huo et al. , 2017; Luo et al. , 2018), kaolinite (Razanamalala et al. , 2017), gibbsite (Razanamalala et al. , 2017), Al and Fe oxides (Razanamalala et al. , 2017; Finley et al. , 2018; Perveen et al. , 2019), POM-C (Perveen et al. , 2019)	exchangeable calcium (Ca)-C, macroaggregate-C, microaggregate-C

[Comment 3] It is always challenging in an analysis of a large number of highly

inter-correlated variables to demonstrate which the key variables are, and to demonstrate cause and effect. The analysis could obviously change if new variables were added (more information on the quality of plant inputs, for example), but I found the analysis presented in Figure 2 to be very effective and convincing. However, I would like to request a couple of further analyses be carried out to demonstrate that the results are robust:

1. Firstly, the priming effects are presented per unit soil weight. Priming effects can also be presented as a percentage stimulation of the flux, per unit SOM, or even per unit C added. It would be helpful to demonstrate that the conclusions hold irrespective of the units the priming effects are presented in and are not simply related to SOM quality and availability because they are presented per unit soil mass.

[Response] Very good comment! Following the reviewer's comment, **we have also presented the priming effect in other three units and re-analysed the data. The results of re-analysis confirmed that SOM stability explained much more variance in relative priming effect than plant, soil and microbial properties** (Fig. R1).

Despite the consistency, we prefer not to use per unit SOC and per unit C added as the unit of priming effect due to the following reason: the purpose of our study was not to compare the magnitude of priming effect among sites, but rather to quantify the relative importance of different factors (*i.e.*, plant, soil, and microbial properties, and SOM stability) in regulating the priming effect over the broad geographic scale. **To achieve this aim, we need to include as many potential drivers as possible in the multivariate analysis.** The SOC content (Kuzyakov, 2000) and the amount of input C (Xiao *et al.*, 2015) are exactly two important variables that have been proved to regulate the priming effects. Thus, we have included these two variables as examined factor in soil (*e.g.*, **SOC**, TN, pH...) and plant (*e.g.*, ANPP, **the amount of input C...**) properties. However, **if we used per unit SOC and per unit C added as the unit of priming effect, we could not evaluate their relationship with priming effect due to the autocorrelation and could not further compare**

their relative importance with other factors either. Probably, this is also the reason why the priming effect was presented in per unit soil dry weight in the recent two large-scale empirical studies (Razanamalala *et al.*, 2017, Perveen *et al.*, 2019). Due to this point and the consistent results derived from various units, we prefer to only use per unit soil weight and percentage of stimulation of flux as unit of priming effect in the revised MS. Nevertheless, we can add the results using per unit SOM or C added if the reviewer persists his/her opinion. Thanks for your understanding!

Figure R1. Relative contributions of SOM stability, and plant, soil, and microbial properties in driving priming effect represented in per unit soil weight (a), percentage of CO₂ flux (b), per unit soil organic C (c) and per unit added C (d). In panel (c), SOC and total N content were removed from the soil variables due to the autocorrelation; in panel (d), the amount of C input, ANPP and total PLFAs were removed from the plant and microbial properties due to the autocorrelation and collinearity.

[Comment 4] 2. In terms of the nutrient mining hypothesis, I was surprised that potential enzyme activity ratios did not seem to be included as microbial predictors. The ratios of C:N, C:P and N:P cycle enzymes may provide information on the extent of microbial C versus nutrient limitation and this would seem more likely to be informative than the raw potential activity measurements.

[Response] Following the reviewer's comment, we have determined the activity of phosphatase, calculated the ratio of C:N, C:P and N:P cycle enzymes, and analyzed their effects on the priming effect. Our additional analyses showed that the priming effect did not exhibit any significant correlations with either enzyme activities (Fig. R2a-c) or the ratios of enzymes (Fig. R2d-f). These results have been incorporated into the Figure 2 of the revised MS (Page 39).

Figure R2. Relationship between the priming effect and C-acquiring enzyme (BG) (a), N-acquiring enzyme (NAG+LAP) (b), P-acquiring enzyme (AP) (c), C:N acquisition ratio (BG/(NAG+LAP)) (d), N:P acquisition ratio ((NAG+LAP)/AP) (e) and C:P

acquisition ratio (BG/AP) (f). The dashed line indicates an insignificant relationship between the two variables. The orange dots and green squares represent data derived from alpine steppe and alpine meadow, respectively.

[Comment 5] 3. The rate of addition was made proportional to initial microbial biomass. I recognise that some kind of standardisation is necessary and using microbial biomass makes sense. However, by standardising per unit microbial biomass this may already reduce the potential importance of microbial variables in controlling the magnitude of priming effects. This should be discussed in the paper.

[Response] Very good comment! We agree with the reviewer that this glucose addition may affect the role of microbial variables in regulating the priming effect, **and discussed this potential uncertainty in the revised MS as follows:** “Although the addition of an amount of glucose equal to 100% of the microbial biomass had been widely adopted in priming experiments (Bastida *et al.*, 2013), the method may affect the interpretation of the importance of microbial properties in regulating the priming effect. To minimize this potential impact, we used the per unit soil weight and % of flux as the unit of the priming effect instead of using the per unit C added to avoid the removing the C input effect. Moreover, we used the microbial community structure (*i.e.*, the ratio of fungi to bacteria, F/B), microbial enzyme activity (*i.e.*, β -1,4-glucosidase, BG; β -1,4-N-acetylglucosaminidase, NAG; leucine aminopeptidase, LAP; and acid/alkaline phosphatase, AP) and their stoichiometric ratios as additional microbial properties.” (Page 20, line 412-420). Thanks for your understanding!

[Comment 6] In summary, I found this to be an interesting and potentially important study but would suggest that the authors need to carry out a few more analyses to demonstrate the robustness of the conclusions and would advise against using the term ‘traditional’ and suggesting that only plant and microbial variables have been investigated to date. Rather, the study would be better framed as a detailed evaluation of the relative importance of plant, microbial and SOM variables in controlling the

magnitude of priming effects.

[**Response**] Following the reviewer’s comments, we have made the following two major changes:

- Following the suggestions from this reviewer and the Reviewer #3, **we have reanalyzed our data in four types of units** (*i.e.*, per unit soil weight, percentage of CO₂ flux, per unit SOC and per unit input C) and **included more variables for plant** (NPP, root biomass, C input amount, relative coverage of grass, forb and sedge), **soil** (silt, sand, bulk density and soil order) **and microbial properties** (P-acquire enzyme activity, the ratios of C:N, C:P and N:P cycle enzymes) to demonstrate the robustness of our conclusion (Table R2). Our additional analyses confirmed that SOM stability was a key factor regulating the priming effect over broad geographic scale.
- **We have deleted the term ‘traditional’ throughout the revised MS, and re-organized our Introduction section to focus more on quantifying the relative importance of different factors in driving the priming effect over broad geographic scale** (Page 3-5, line 66-100).

Table R2. Variables added to represent plant, soil and microbial properties in the original and revised MS.

Factor	Variables used in original version of the MS	New variables added in the revised MS
Plant properties	ANPP and EVI	NPP, root biomass, C input amount, relative coverage of grass, forb and sedge
Soil properties	SOC, pH and clay	silt, sand, bulk density and soil order
Microbial properties	Total PLFA, bacterial PLFA, F/B ratio, C-acquiring enzyme and N-acquiring enzyme	P-acquiring enzyme, C:N ratio of enzyme, N:P ratio of enzyme and C:P ratio of enzyme

[Comment 7] Specific comments: Lines 277-315: How frequently were the flux and isotopic measurements made and did the magnitude or sign of the priming effects change with time during the 65 day incubation?

[Response] We determined the flux and isotopic signal on six occasions, *i.e.*, days 1, 3, 8, 15, 35 and 65. We have clearly mentioned this point in the revised MS (Page 16, line 331-333). Our results showed that the magnitude of priming effect decreased exponentially with incubation time (Fig. R3). However, the sign of priming effect did not change in 73.3% of study sites, but shifted from positive to negative priming in 26.7% of study sites. Notably, changes in the sign and magnitude of priming effect have been frequently observed in other priming studies (Thiessen *et al.*, 2013, Qiao *et al.*, 2014, Ye *et al.*, 2015).

Figure R3. Changes in the sign and magnitude of the priming effect with the incubation time for soil samples collected from 30 study sites. The red dotted line indicates the zero line.

[Comment 8] Lines 326-327: Are there no data on plant species composition. *E.g.*

relative coverage of grasses, sedges, shrubs etc?

[Response] We have data on the relative coverage of grass, forb and sedge, and analyzed their effects on the priming effect. Our additional analyses showed that the priming effect did not exhibit any significant correlations with the coverage of grass and sedge (Fig. R4a, c), but was positively related to the coverage of forb (Fig. R4b). These results have been incorporated into the Fig. 2 (Page 39). It should be noted that shrub is not the dominant group in Tibetan alpine grassland (Editorial Committee for Vegetation Map of China, 2001), so we don't have the data on shrub.

Figure R4. Relationships of priming effect with the relative coverage of grass (a), forb (b) and sedge (c). The dashed line indicates an insignificant relationship between the two variables, while the solid line represents the fitted ordinary least squares model, with the grey area corresponding to 95% confidence intervals. * denotes significant correlation between priming effect and the proportion of fast C pool at $P < 0.05$. The orange dots and green squares represent data derived from alpine steppe and alpine meadow, respectively.

[Comment 9] Lines 342-347: The ratios of potential enzyme activities associated with carbon, nitrogen and phosphorus cycles may provide information on how carbon or nutrient limited the microbes were. In terms of testing hypotheses related to nutrient mining, these ratios may be more valuable than individual enzyme activities.

[Response] As mentioned above, we have included the ratio of enzyme activities as microbial properties in the revised MS (Page 20-21, line 425-427).

Taken together, we appreciate for the reviewer's insightful comments. These comments enabled us to have a deeper thinking on data analysis, and the major point delivered by the current study. By addressing these comments, we feel that the revised MS has been greatly improved. Thank you!

Response to Reviewer #2:

[Comment 1] Chen et al. evaluated soil priming across a gradient of sites in the Tibetan Plateau. I acknowledge the amount of work that went into this project. However, I had a difficult time with the manuscript for the reasons given below.

To start with, I don't completely accept the justification given several times, that "none of previous studies has explicitly addressed the relationship between soil C decomposability and priming effect over the broad geographic scale" (lines 74-76). There are numerous related gradient studies (e.g., Madagascar, <https://www.nature.com/articles/ismej2017178>). Why are these studies not relevant?

[Response] We are very sorry for neglecting the relevant references. Following the reviewer's comment, we have carefully consulted the relevant literatures and cited them in the revised MS. Based on this literature review, we find that there are three major differences between our study and other similar studies:

(1) While data synthesis and empirical evidence provide basic understandings on potential drivers of priming effect (Luo *et al.*, 2015, Luo *et al.*, 2016, Huo *et al.*, 2017, Razanamalala *et al.*, 2017, Perveen *et al.*, 2019), **as mentioned by the Reviewer #1, the relative importance of different factors (i.e., plant, soil, and microbial properties and SOM stability) in governing the priming effect has not yet been quantified. To fill this knowledge gap, this study combined three types of statistical analysis (i.e., partial correlation, variation partitioning analysis, and structural equation modelling) to quantify the relative importance of different factors in driving regional patterns of priming effect.**

(2) Despite increasing evidence highlights the vital role of SOM stabilization mechanisms (i.e., chemical recalcitrance and physico-chemical protection) in regulating soil C turnover (Schmidt *et al.*, 2011, Lehmann & Kleber, 2015, Bradford *et al.*, 2016), **as mentioned by Reviewer #3, it remains unclear about the role of these mechanisms (hereafter as 'SOM stability') in regulating the priming effect over broad geographic scale** (Razanamalala *et al.*, 2017). To be specific, previous studies characterized SOM quality indirectly by using C:N ratio (Luo *et al.*, 2016,

Qiao *et al.*, 2016, Razanamalala *et al.*, 2017) or DOC (Chowdhury *et al.*, 2014) and thus **it remains unknown whether and how different SOM compositions** (*e.g.*, carbohydrate, lipid, and lignin), which could represent chemical recalcitrance more directly, **regulate the priming effect over the large scale. Due to this point, our study included more variables to directly characterize chemical recalcitrance** (*e.g.*, carbohydrate, lipid, lignin...), **and then explored their effects on priming effect by combining acid hydrolysis, biomarker analysis and two-pool C decomposition model to explore the relationship between the priming effect and different SOM fractions.**

(3) Recent studies have reported the dependency of priming effect on clay content (Huo *et al.* 2017; Luo *et al.* 2018), aluminum (Al) and iron (Fe) oxides (Razanamalala *et al.*, 2017, Finley *et al.*, 2018), **but it remains unclear about the relationships between priming effect and calcium (Ca) and aggregate-mediated protection over broad geographic scale.** Given that the protection mediated by Ca²⁺ and soil aggregate may outweigh Al/Fe oxides in neutral and alkaline soil environments (Rowley *et al.*, 2018), our understanding on the drivers of priming effect will be incomplete without considering their effects. **Due to this point, in addition to those variables used in previous studies** (*e.g.*, clay content, Al and Fe oxides), **this study further explored the effect of Ca- and aggregate-mediated stabilization on the priming intensity by combining aggregate fraction and mineral analysis to further explore the relationship between the priming effect and physico-chemical protection.**

Based on these new understandings, we have deleted the related sentence: “*none of previous studies has explicitly addressed the relationship between soil C decomposability and priming effect over the broad geographic scale*”, and re-organized the *Introduction* section to accurately describe our novelty in the revised MS (Page 3-5, line 66-100). Thanks for your understanding!

[**Comment 2**] *More importantly, the paper doesn't do a good job of explaining the 2200-km gradient, either its characteristics or how it relates to the study system. The gradient is described in this way: "The sampling sites covered a wide range of climate, plant productivity and soil conditions affecting soil C stock and its decomposability." Why and how were these 90 sites chosen? How do soil properties change along this gradient, in particular texture and other physical properties. Why, for instance, would you necessarily incubate all soils with fixed soil moisture (60% water holding capacity) (line 280) if there are textural, and hence water potential, differences? It feels at times that the authors were designing an experiment to reach a particular conclusion.*

[**Response**] Very good comment! Before addressing your comment, **we would like to mention that the 90 samples used in this incubation were collected from 30 sites (i.e., three replicates from each site) rather than 90 sites. To avoid the confusion, we have rephrased the related sentence** as follows: "In this study, we quantified the relative importance of various factors (*i.e.*, plant, soil and microbial properties and SOM stability) in regulating the priming effects based on 30 sites along an approximately 2200 km grassland transect on the Tibetan Plateau" (Page 5, Line 102-104). We would then address the reviewer's comments from the following two aspects:

- Regarding to the site selection, **the selection criteria is that the environmental gradients** (*i.e.*, climate, plant, soil and microbial properties) **involved in these sites should be as large as possible,** since this study aimed to quantify the relative importance of different factors (*i.e.*, plant, soil, and microbial properties, and SOM stability) in driving the regional pattern of priming effect. As reported in previous studies (Yang *et al.*, 2009, Chen *et al.*, 2016d, Ding *et al.*, 2016b), **the Tibetan Plateau is the highest and largest plateau in the world and has a broad environmental gradient** (Yang *et al.*, 2009); **as such, this area serves as an ideal platform for exploring the dominant drivers of the priming effect over a broad geographic scale.** Specifically, the climate is characterized as cold

and dry across the plateau, with a southeast to northwest precipitation gradient ranging from 84 to 593 mm per year. The mean annual temperature in this area varies from -4.9 to 6.9 °C (Ding *et al.*, 2016b). The dominant ecosystem on the plateau is alpine grassland, which shifts from alpine steppe in the northwestern area to alpine meadow in the southeastern area (Yang *et al.*, 2009). The soil orders in this region include Cambisol, Calcisol, Chernozem, and Kastanozem according to the World Reference Base for Soil Resources (Shi *et al.*, 2004). To cover this broad environmental gradient, we sampled 30 sites (including 18 steppe sites and 12 meadow sites) along an approximately 2200 km grassland transect during the growing seasons (between early July and early September) in 2013 and 2014 (Fig. R5). These sites were set throughout the geographical extent of alpine grasslands, **covering a wide range of climates (mean annual temperature: -3.7–6.9 °C, mean annual precipitation: 89–534 mm), plant productivity levels (net primary productivity (NPP): 37.9–488.3 g m⁻² yr⁻¹), soil properties (organic C: 1.1–118.3 g kg⁻¹; pH: 6.2–9.4; silt content: 5.0–51.8%; bulk density: 1.0–1.5 g cm⁻³) and microbial properties (microbial biomass C: 22.8–1101.2 mg C kg⁻¹)** across the plateau (Chen *et al.*, 2016c, Ding *et al.*, 2016b) (Fig. R6). We have added this background information and selection criteria in the revised MS (Page 13-14, line 275-299).

- Regarding to the fixed soil moisture, we would like to mention that **adjusting soil samples to the same percentage of water holding capacity allows us to eliminate the potential weather impact on soil moisture during field sample collection** (Colman & Schimel, 2013). This is particularly important for large-scale laboratory incubations with soil samples from different sites (Fierer *et al.*, 2006, Craine *et al.*, 2010, Colman & Schimel, 2013, Ding *et al.*, 2016a, Razanamalala *et al.*, 2017). Moreover, **given that ~60% WHC is the optimal water potential for microbial growth** (Howard & Howard, 1993, Rey & Jarvis, 2006), **fixing the soil moisture to the same water holding capacity percentage could also facilitate the comparison of microbial processes across different**

sites (Fierer *et al.*, 2006). For these reasons, this procedure has been widely used in previous studies regarding SOM decomposition and priming effect (Fig. R7) (Fierer *et al.*, 2006, Craine *et al.*, 2010, Colman & Schimel, 2013, Ding *et al.*, 2016a, Razanamalala *et al.*, 2017). For example, soil moisture was also set to a fixed percentage of water holding capacity in the recent large-scale priming study mentioned by this reviewer (Razanamalala *et al.*, 2017). In addition, the gravimetric water content during the laboratory incubation still had large variations from 22.7~98.2% after adjustment, and was also closely correlated with the corresponding value during the field sampling ($r^2 = 0.81$, $P < 0.01$; Fig. R8), reflecting that there was still a spatial gradient in soil moisture after the adjustment. We have clearly mentioned these points in the *Methods* section of the revised MS (Page 15-16, line 316-322). Thanks for your understanding!

Figure R5. Distribution of the sampling sites in alpine grasslands on the Tibetan Plateau.

Figure R6. Bubble plot representing the spatial patterns of climate, plant, soil and microbial properties. MAT (a), MAP (b), NPP (c), ANPP (d), MBC (e), SOC (f), BD (g), pH (h), clay (i) and silt (j). MAT, mean annual temperature; MAP, mean annual precipitation, NPP, net primary productivity; ANPP, aboveground net primary productivity; MBC, microbial biomass carbon; SOC, soil organic carbon; BD, bulk density. In panel (a), black bubble indicates the negative value, while red bubble indicates the positive value.

Figure R7. The percentage of different procedures used to adjust soil moisture in priming experiments.

Figure R8. Relationship between incubation moisture and field moisture across 30 study sites along a grassland transect on the Tibetan Plateau. The solid line represents the fitted ordinary least squares model, with the grey area corresponding to 95% confidence intervals. *** denotes significance at $P < 0.001$. The orange dots and green squares represent data derived from alpine steppe and alpine meadow, respectively.

[Comment 3] The words “error” and “statistics” are barely mentioned in the

manuscript and not at all in the supplement. Much more work and description is needed here for the paper to be acceptable.

[Response] Thanks for the reviewer's insightful comments! **Following the reviewer's comment, we used the Monte Carlo approach to estimate the 95% confidence interval of the cumulative priming intensity and the relative priming effect** (Zhu & Cheng, 2011, Luo *et al.*, 2015). Specifically, we first quantified the uncertainties in the SOM-derived CO₂ release for each time's CO₂ and δ¹³C measurements under both the glucose treatment and control conditions. The mean values and standard deviations (SDs) of CO₂ release under the control conditions were calculated based on triplicate samples. In contrast, the SD of SOM-derived CO₂ release under the glucose treatment was calculated according to the following error propagation in Eq. (1) (Ku, 1996; Qiao *et al.*, 2014), in which the uncertainties in both the CO₂ flux and δ¹³C measurements were considered simultaneously.

$$\sigma_{SOM} = \sqrt{\left(\frac{\sigma_{C_{treat}}}{C_{treat}}\right)^2 + \left(\frac{\sigma_{f_{SOM}}}{f_{SOM}}\right)^2} \quad (1)$$

where σ_{SOM} is the SD of the SOM-derived CO₂ flux in the glucose treatment, $\sigma_{C_{treat}}$ and $\sigma_{f_{SOM}}$ are the SDs of the total CO₂ fluxes in the glucose-treated soil and the SOM pool fraction, respectively, and C_{treat} and f_{SOM} are their respective mean values.

After obtaining the means and SDs of SOM-derived CO₂ release under both the glucose treatment and control conditions, we performed 1000 Monte Carlo simulations for the priming effect during each time's measurement. First, the SOM-derived CO₂ releases in both the glucose treatment and control conditions were randomly generated based on normal distributions using the abovementioned means and SDs. The priming effect was then calculated as the difference in the SOM-derived CO₂ release between the glucose treatment and the control. The cumulative priming effect was further estimated by integrating the absolute priming effect over the incubation time. Finally, we calculated the 95% confidence interval of the cumulative

priming effect based on a total of 1000 estimates derived for each study site. We have added the detailed information on the uncertainty estimation in the *Methods* section of the revised MS and also added these errors in supplementary Table 1 (Table R3; Page 17-19, line 362-386). Additionally, we also have clearly presented the related statistical methods and significant analysis (*P* value) in the *Results* session of the revised MS (Page 6-8, line 133-158).

Table R3. The background information and the priming effect for the 30 sampling sites.

Site No.	Altitude (m)	MAT (°C)	MAP (mm)	Vegetation type	Dominant species	Soil order	Priming effect (µg /g soil dry weight)	Relative priming effect (%)
1	3112	5.08	322	AS	Stipa sareptana	Chernozems	96.5 ± 6.3	28.5 ± 1.9
2	3280	2.02	369	AS	S. sareptana	Chernozems	66.3 ± 9.9	20.3 ± 3.0
3	3650	-0.43	496	AM	Kobresia pygmaea	Cambisols	34.0 ± 10.6	8.0 ± 2.5
4	3830	-0.84	531	AM	K. pygmaea	Cambisols	110.5 ± 12.9	37.3 ± 4.1
5	4032	-1.34	534	AM	Elymus nutans	Cambisols	112.2 ± 9.8	37.3 ± 3.2
6	4417	-2.93	419	AM	K. tibetica	Cambisols	37.1 ± 8.4	6.5 ± 1.4
7	4228	-3.25	321	AS	S. purpurea	Calcisols	57.1 ± 6.8	15.6 ± 1.9
8	4430	-4.09	508	AS	S. purpurea	Cambisols	52.2 ± 5.4	17.9 ± 1.6
9	4162	-0.39	495	AM	K. pygmaea	Cambisols	55.5 ± 10.6	13.1 ± 2.2
10	4289	0.15	481	AM	K. pygmaea	Cambisols	89.6 ± 9.2	18.2 ± 1.9
11	4161	-1.76	415	AM	K. pygmaea	Cambisols	51.7 ± 5.1	10.1 ± 1.0
12	4165	-0.55	338	AS	S. purpurea	Cambisols	91.6 ± 4.6	26.5 ± 1.3
13	3224	2.26	324	AS	S. purpurea	Cambisols	76.2 ± 7.2	16.3 ± 1.6
14	4637	-4.15	295	AS	Carex moorcroftii	Calcisols	15.9 ± 4.5	5.3 ± 1.5
15	3137	1.57	314	AS	S. breviflora	Kastanozems	113.8 ± 11.7	26.2 ± 2.7
16	3262	0.06	242	AS	S. sareptana	Kastanozems	132.5 ± 17.2	30.5 ± 4.0
17	4544	-2.69	89	AS	C. moorcroftii	Calcisols	69.5 ± 17.1	13.7 ± 3.4
18	4622	-3.26	307	AM	K. pygmaea	Calcisols	56.7 ± 4.5	11.9 ± 1.0
19	4687	-2.84	307	AM	Festuca ovina	Calcisols	78.0 ± 5.0	14.9 ± 1.0

20	4639	-1.80	503	AS	S. purpurea	Calcisols	63.9 ± 10.2	17.6 ± 2.7
21	4298	-0.35	515	AM	K. pygmaea	Cambisols	42.9 ± 5.8	7.9 ± 1.1
22	4748	-0.65	332	AS	C. moorcroftii	Calcisols	84.3 ± 4.8	19.1 ± 1.1
23	4590	-0.96	337	AS	S. purpurea	Calcisols	28.3 ± 4.4	5.4 ± 0.9
24	4544	0.32	273	AS	S. purpurea	Calcisols	67.3 ± 4.0	15.4 ± 0.9
25	4560	0.43	209	AS	Deyeuxia arundinacea	Calcisols	26.4 ± 3.2	8.7 ± 1.0
26	4444	0.5	170	AS	S. caucasica	Calcisols	43.9 ± 5.7	9.6 ± 1.2
27	4390	0.91	127	AS	S. tianschanica	Calcisols	46.0 ± 13.6	16.2 ± 5.0
28	4538	1.12	99	AS	S. glareosa	Calcisols	-3.95 ± 0.6	-1.1 ± 0.1
29	4294	1.89	464	AM	K. pygmaea	Cambisols	100.5 ± 6.7	18.6 ± 1.2
30	4764	0.08	457	AM	K. pygmaea	Cambisols	77.7 ± 4.7	14.2 ± 0.9

Note: MAT, mean annual temperature; MAP, mean annual precipitation; AS, alpine steppe; AM, alpine meadow. The soil order was derived from the 1:1,000,000 digital soil map of China (Shi *et al.*, 2004). The data of priming effect are represented as mean ± 95% confidence interval estimated by the Monte Carlo method.

[Comment 4] The writing will need considerable work for grammar and clarity before publication, regardless of the outcome of review. That statement is not a reflection on the review or a factor to weigh for acceptance or rejection.

[Response] Following the reviewer's comment, we have asked an English language editing service (*i.e.*, Springer Nature Author Services) for language check. Please see the certification at the end of this response letter.

[Comment 5] I don't find Figure 2 to be very helpful. It is difficult to read and somewhat difficult to interpret.

[Response] Sorry for the confusion. Figure 2 showed the results of partial correlation, which was one of the multivariate analyses (*i.e.*, partial correlation, variation partitioning analysis, and structural equation modelling) we used to explore the relative importance of individual factors. By applying the partial correlation, we can evaluate the strength of linear associations between priming effect and other factors that cannot be considered by the controlled predictor (Doetterl *et al.*, 2015). For example, our results indicated that, in the absence of controlling the role of SOM stability (zero-order in Fig. 2a), the priming intensity was significantly correlated with plant, soil environment and microbial properties. However, after controlling C decomposability, the correlation coefficients between priming effect and plant, soil and microbial predictors decreased by 63.5%, 69.2% and 92.4%, respectively, so that most of these factors were no longer associated with the priming effect (Fig. 2a). In contrast, no sharp decreases were observed in correlation coefficients between the priming intensity and C decomposability after controlling the plant, soil environment and microbial factors (Fig. 2b). Here you can see that, this partial correlation analysis provided another evidence that SOM stability was the dominant factor modulating the priming intensity over the large scale. That's also why the Reviewer 1# mentioned that: "*I found the analysis presented in Figure 2 to be very effective and convincing*". In addition, this type of heat map has been frequently used to visualize the results (*e.g.*, the (partial) correlation coefficient, response ratio, RMSE and so on) in previous studies (Doetterl *et al.*, 2015, Peng *et al.*, 2015, Tian *et al.*, 2015, Luo *et*

al., 2017, Mooshammer *et al.*, 2017, Gentsch *et al.*, 2018).

Based on above-mentioned aspects, we prefer to keep this figure in the revised MS.

To improve its readability, we have added the detailed interpretations in the figure legend of the revised MS as follows:

“The x-axis shows the zero-order (without controlling any factors) and the factors being controlled. The y-axis shows the factors of which the correlations with the priming effect are examined. The size and colour of the circles indicate the strength and sign of the correlation. Differences in circle size and colour between the zero-order and controlled factors indicate the level of dependency of the correlation between the priming effect and the examined factor on the controlled variable (no change in circle size and colour between the controlled factor and zero-order = no dependency; a decrease/increase in circle size and colour intensity = loss /gain of correlation)” (Page 39-40, line 766-771).

Nevertheless, we could delete this figure if the reviewer persists his/her opinion.

Thanks for your understanding!

[Comment 6] *Some relevant papers:*

Meta-analysis:

<https://www.sciencedirect.com/science/article/pii/S0038071716301560>

Priming in a similar steppe in China:

<https://onlinelibrary.wiley.com/doi/full/10.1111/oik.01728>

[Response] Many thanks for these relevant papers. We have carefully read them and cited them in the revised MS (Page 4, line 68, 76, 91).

[Comment 7] *Additional comments: It's my understanding that the authors split all predictors into four groups: plant factors, soil factors, microbial factors, and soil C decomposability. It seems that a few measurements are missing from the first three groups, and the uniqueness of soil C decomposability as a predictor is unclear.*

For the 'plant factor', we know belowground inputs (e.g., root biomass and exudation) play a key role in SOM formation and in situ priming. While the transect was

dominated by similar vegetation (namely, steppe and meadow grasslands), would you expect significant differences in below-ground biomass across sites? For example, different root biomass and/or rooting depths, depending on climate. Do grass species change across the transect?

[Response] Very good comment! Following the reviewer’s comment, we have included the data of root biomass, relative coverage of grass, forb and sedge as additional ‘plant predictors’ (Table R4). We also added the information of dominant species in the in supplementary Table 1 (Table R3). These plant variables exhibited large variations among the 30 study sites, with root biomass, the relative coverage of grass, forb and sedge ranging from 31-1954 g m⁻², 0.5-95.1%, 0.6-78.8% and 0-98.9%, respectively (Fig. R9). Moreover, the priming intensity was positively correlated with both root biomass ($r^2 = 0.18$, $P < 0.05$; Fig. R9a) and the relative coverage of forb ($r^2 = 0.28$, $P < 0.01$; Fig. R9c), but was independent on the relative coverage of grass and sedge. We have incorporated these data and associated analyses into the Figure 2 of the revised MS (Page 39).

Frankly speaking, it is still a big challenge to obtain the data of rooting depth and root exudation over broad geographic scale (Phillips *et al.*, 2008), although we do agree that both of them may have important effects on the priming effect in the field. Nevertheless, the variation in rooting depth across sites might not affect our conclusions because we only focused on the topsoil (0-10 cm) priming effect. Additionally, the amount of root exudation could be partly reflected by root biomass and NPP (Chapin *et al.*, 2012). Thanks for your understanding!

Table R4. Variables added to characterize the plant, soil and microorganism predictors in the original and revised MS.

Predictor	Variables used in the original MS	New variables added in the revised MS
Plant	ANPP and EVI	NPP, root biomass, C input amount, relative coverage of grass,

		forb and sedge
Soil	SOC, pH and clay	silt, sand, bulk density and soil order
Microorganism	Total PLFA, bacterial PLFA, F/B ratio, C-acquiring enzyme and N-acquiring enzyme	P-acquiring enzyme, C:N ratio of enzyme, N:P ratio of enzyme and C:P ratio of enzyme

Figure R9. Relationships of the priming effect with plant variables including root biomass (a), and the relative coverage of grass (b), forb (c) and sedge (d). The dashed line indicates an insignificant relationship between the two variables, while the solid line represents the fitted ordinary least squares model, with the grey area corresponding to 95% confidence intervals. * and ** denotes significance at $P < 0.05$ and $P < 0.01$. The orange dots and green squares represent data derived from alpine steppe and alpine meadow, respectively.

[Comment 9] The 'soil factor' does not include any information on soil mineralogy or other inherent soil characteristics. I would recommend including soil texture (% clay and silt) and soil order, at the very least, as a part of this factor. The authors could also include bulk density.

[Response] Very good comment! Following the reviewer's comment, we have incorporated soil texture (clay, silt and sand), soil order (Chernozems, Cambisols, Calcisols and Kastanozems) and bulk density data as additional soil properties (Table R4; Page 24-25, line 509-510). Our additional analysis showed that the priming effect was negatively correlated with bulk density ($r^2 = 0.16$, $P < 0.05$), but was not associated with soil texture (Fig. R10). Moreover, the priming effect varied significantly among soil types (Fig. R11), with highest priming intensity in Kastanozems and lowest in Calcisols. Additionally, the updated multivariate analyses after adding all these new plant and soil variables, confirmed that SOM stability explained more variance in priming effect than plant, soil and microbial properties (Fig. R12). Notably, we do agree that soil mineralogy could play an important role, but unfortunately, we did not determine it in current study. Therefore, future studies should give more attention on its potential role in regulating priming effect. Thanks for your understanding!

Figure R10. Relationship between priming effect and additional soil variables including clay content (a), silt content (b), sand content (c) and bulk density (d). The dashed line indicates an insignificant relationship between the two variables, while the solid line represents the fitted ordinary least squares model, with the grey area corresponding to 95% confidence intervals. * denotes significance at $P < 0.05$. The orange dots and green squares represent data derived from alpine steppe and alpine meadow, respectively.

Figure R11. Differences in the priming effect among four soil orders. The soil order was derived from the 1:1,000,000 digital soil map of China (Shi *et al.*, 2004)

Figure R12. Relative contribution of SOM stability, plant, and soil and microbial properties in driving priming effect. Variation partitioning analysis was conducted to identify the variance of priming effect using the original dataset (a) and new dataset including additional plant, soil and microbial properties (b) listed in Table R4.

[Comment 10] With 'soil C decomposability' as a combination of chemical quality (e.g., lignin, etc.) and mineral protection, it seems that it would in fact be driven by underlying soil, plant, and microbial factors. The uniqueness and usefulness of considering this as a novel 'non-traditional factor' for analyzing priming is unclear to me.

[Response] Very good comment! We agree with the reviewer that 'SOM stability' has significant interactions with plant, soil and microbial properties, especially it is closely related to soil variables. Nevertheless, we would like to mention that **soil properties** such as the SOC, total N, pH, soil texture and bulk density **reflect more for basic soil environmental conditions. In contrast, the 'SOM stability' as a combination of chemical composition** (*i.e.*, relative abundance of carbohydrate, cutin, suberin and lignin...) **and physico-chemical protection** (*i.e.*, the degree of SOM protection by Al/Fe oxides, exchangeable Ca and aggregates) **provides more valuable insight into the features of soil organic matter (SOM) stabilization**

(Lehmann & Kleber, 2015), **which can regulate the priming effect more fundamentally and directly**. Despite of its importance, **empirical evidence for the effects of SOM stability on the priming effect over the large scale is scarce** (but see (Razanamalala *et al.*, 2017)). **Especially, it remains unknown about the relative importance of SOM stability compared with other factors** (*i.e.*, plant, soil and microbial properties) **in regulating the priming intensity**.

By applying multivariate statistical analysis (*i.e.*, partial correlation, variation analysis and structure equation modelling), **our study demonstrated that ‘SOM stability’ explained more variance in priming effect than plant, soil and microbial properties. This finding emphasizes the importance of considering the molecular composition and mineral protection of SOM in Earth System Models to predict soil C dynamics under changing plant C inputs.** That’s also why Reviewer 3# pointed out that: “*The manuscript is an interesting attempt to estimate the importance of different carbon stabilization process (chemical recalcitrance and mineral interaction)*”. **For these reasons, we feel that the effects of ‘SOM stability’ on the priming effect are different from those of above-mentioned soil variables, and should be considered separately.** Thanks for your understanding!

[Comment 11] Other questions/comments: - What were the decay rates obtained for the two-pool model? Was a unique solution obtained, or what was the distribution? Was there an uncertainty in the size of the fast and slow pools? How did this, and other measurement uncertainties, follow through the later statistical analyses?

[Response] Thanks for these valuable comments. Before addressing your comments, we would like to explain the modeling approach used in this study. Specifically, we can estimate the decay rate (*i.e.*, k_1 and k_2) and the relative proportion of fast and slow pool (*i.e.*, f_1 , $1-f_1$) from the two-pool model. These parameters were determined by the Markov Chain Monte Carlo (MCMC) approach based on Bayesian probabilistic inversion (Xu *et al.*, 2006, Liang *et al.*, 2015). Therefore, **all the estimated parameters (*i.e.*, k_1 , k_2 and f_1) had their distributions with uncertainty** (Fig. R13).

We would also like to emphasize that, only f_1 was used for data analysis among all these parameters estimated from the two-pool model. Moreover, f_1 was also only used for regression analysis as a supplementary evidence (Fig. S2b in original version of the MS) for the observed relationship between the priming intensity and chemical recalcitrance (Fig. R14). Particularly, it was not used for the subsequent multivariate statistical analysis (i.e., partial correlation, variation partitioning analysis and structure equation modelling). Due to this point, we think that this parameter estimation based on modeling approach will not generate too much errors into subsequent analyses. Thanks for your understanding!

Site 15 (Alpine steppe)

Site 19 (Alpine meadow)

Figure R13. Frequency distribution of estimated parameters (k_1 , k_2 and f_1) at a typical alpine steppe site (a-c) and alpine meadow site (d-f).

(a) Estimated data from two-pool model

(b) Measured data from acid hydrolysis and biomarker analysis

Figure R14. Relationship between the priming effect and chemical recalcitrance variables derived from two-pool model (a) and from acid hydrolysis and biomarker analysis (b). The solid line represents the fitted ordinary least squares model, with the grey area corresponding to 95% confidence intervals. * and ** denote significance at $P < 0.05$ and $P < 0.01$. The orange dots and green squares represent data derived from alpine steppe and alpine meadow, respectively.

[Comment 12] Do the priming effect measurements (e.g., Fig. 1) have error associated with them? How much did the incubation results (performed in triplicate) vary?

[Response] Yes, they do. To characterize these errors, we used the Monte Carlo approach to estimate the 95% confidence interval of the cumulative priming intensity (Zhu & Cheng, 2011, Luo *et al.*, 2015) (see details in the response to this reviewer's *[Comment 3]*), in which uncertainties of both CO_2 flux and $\delta^{13}\text{C}$ measurements were taken into consideration simultaneously (Page 17-19, line 362-386). Our estimation indicated that the 95% confidence intervals of the cumulative priming effect were approximately equal to 13.3% of the corresponding means for 30 study sites (Table R3). We have incorporated these results in the Supplementary Table 1.

[Comment 13]- How did you decide to add the ^{13}C glucose equivalent of 100% of the microbial biomass at each site, as opposed to a different proportion or an addition weighted on the relative amounts of SOC or plant inputs across the sites?

[Response] We would like to mention that the purpose of adding the ^{13}C glucose

equivalent to 100% of microbial biomass was to roughly simulate the amount of plant C input in situ, given the strong dependence of MBC on plant C input ($r^2 = 0.64$, $P < 0.01$; Fig. R15a). We agree that it is best to add glucose weighted on the actual amount of plant C inputs under field conditions. However, it is still challenge to accurately estimate the amount of plant C inputs, especially for multiple study sites. Therefore, **the amount of C addition was usually based on SOC** (Kuzyakov & Bol, 2006, Hopkins *et al.*, 2014, Qiao *et al.*, 2016) **or microbial biomass C (MBC)** (Blagodatskaya *et al.*, 2007, Bastida *et al.*, 2013, Sullivan & Hart, 2013) **in most of previous studies.**

In this study, we chose 100% MBC rather than the relative amounts of SOC because the amount of added available substrate in relation to the microbial C has been demonstrated to be a key factor affecting the direction and magnitude of the priming effect (Blagodatskaya & Kuzyakov, 2008). **It was highlighted that studies comparing priming effects among different soils or among different horizons should consider microbial biomass** (Blagodatskaya & Kuzyakov, 2008), **and measuring the quantity of C added to soil relative to the size of the microbial biomass pool is a useful manner of inter-study comparisons of the priming effect** (Sullivan & Hart, 2013). That's also why Reviewer 1# mentioned that "*I recognise that some kind of standardisation is necessary and using microbial biomass makes sense.*" Furthermore, the amount of C input amount was approximately equal to 22% NPP, which was within the natural range (root exudates account for 10-30% of NPP; Chapin *et al.*, 2012). In addition, given the positive correlation between MBC and SOC ($r^2 = 0.66$, $P < 0.01$; Fig. R15b), the addition weighted on the MBC of each site should be similar to the addition weighted on SOC. We have clearly mentioned these points in the *Methods* section of the revised MS (Page 16, line 324-329).

Figure R15. Relationship between microbial biomass carbon and net primary production (NPP, a) and soil organic carbon (b) across our study sites. The orange and green dots represent data derived from alpine steppe and alpine meadow, respectively.

[Comment 14]- Do you expect that a large addition of glucose, a low molecular weight ‘sweet’, could shift the microbial community? Does this limit the generalizability of the results? It seems the glucose addition was performed as a pulse at the beginning of the incubation. Can you comment on the potential role and implications of C starvation during the course of the 65-day experiment?

[Response] We agree with the reviewer that a single pulse addition of glucose may shift microbial community (Chen *et al.*, 2014) and induce C starvation during the course of 65-day incubation (Qiao *et al.*, 2016), although this is a frequently used approach in the priming experiment (Reinsch *et al.*, 2013, Mau *et al.*, 2015, De Baets *et al.*, 2016, Song *et al.*, 2018), especially for the large-scale study (Razanamalala *et al.*, 2017, Perveen *et al.*, 2019). This C starvation can induce the decrease in priming effect with incubation time (Qiao *et al.*, 2016).

Due to this point, **we have discussed this limitation in the Discussion section as follows**: “despite being a frequently used approach in priming experiments, **a single pulse of glucose addition with an amount equal to the microbial biomass cannot realistically characterize plant C inputs in terms of quantity, quality** (Cheng *et al.*, 2014, Qiao *et al.*, 2016) **and input frequency in natural ecosystems. This single**

addition may also induce changes in microbial biomass (Chen *et al.*, 2014), community structure (Fontaine *et al.*, 2003, Hobbie & Hobbie, 2013) and microbial C use efficiency (Wild *et al.*, 2014). Furthermore, **due to** the decrease in microbial diversity (Mau *et al.*, 2015) and **the microbial C starvation during the late stage of incubation** (Qiao *et al.*, 2016), **this single addition may result in lower priming effect or even negative priming compared with repeated substrate addition** (Hamer & Marschner, 2005, Mau *et al.*, 2015). **In situ experiments are thus encouraged to better capture realistic plant C inputs and elucidate the role of plant properties in regulating the priming effect over a large scale**” (Page 12, line 243-255). Thanks for your understanding!

[Comment 15]- The term ‘mineral-affected’ is a bit confusing, and I have not seen it used in the literature. I understand the intent — that the micro-aggregate size fraction is influenced by the formation of underlying mineral-associations and sticky microbial compounds, and therefore they are both in some way ‘mineral-affected’. However, these two fractions are very different mechanistically — namely, through physical vs. chemical protection. I would suggest breaking up this pool into clay+silt vs. micro-aggregate contributions, or I would use a different term.

[Response] Following the reviewer’s suggestion, we have broken up the ‘mineral-affected C’ pool into micro-aggregate and clay + silt, and also added the macro-aggregate contribution in the revised MS (Fig. R16).

Figure R16. Relationships of priming effect with OC proportion included in macro-aggregate (a), micro-aggregate (b) and clay and silt (c). The orange and green

dots represent data derived from alpine steppe and alpine meadow, respectively.

[Comment 16]- How would this information be used to inform models? While priming is recognized to be an important feedback and efforts are being made to capture this response in models, it is still debated how to mechanistically or implicitly incorporate this feedback in models. What do the results tell us about mechanisms and correct representations? How can these reported correlations be used? Furthermore, such SOM models are often applied across soil and vegetation types. Do you expect your results to be generalizable to other vegetation types, e.g., forests?

[Response] Very good comment! Currently, the priming intensity was solely assumed to depend on the amount of plant C input in modelling studies (Perveen *et al.*, 2014, Sulman *et al.*, 2014, Guenet *et al.*, 2018), with SOM properties being seldom considered. However, our results demonstrated that SOM stability explained more variance in priming effect than plant, soil and microbial properties. **The higher predictive power of the SOM stability infers potential uncertainties in predicting the priming effect and subsequent soil C dynamics among current models. To improve model predictions, the conceptual framework of chemical recalcitrance and physico-chemical protection of SOM should be incorporated into Earth System Models. Therefore, our reported correlation could be used to improve the prediction accuracy of priming effect in grassland ecosystems.** Nevertheless, given the distinct differences in soil properties and priming effect across various ecosystems (Luo *et al.*, 2016), more studies covering various vegetation types (*e.g.*, forest, shrubland and desert) are needed to further advance our understanding of the regulatory mechanisms of SOM dynamics. Due to this point, we also discussed this limitation in the *Discussion* section as follows: “In addition, the limited ecosystem types involved in this study may have also induced uncertainties for upscaling. More empirical studies with diverse ecosystem types (forests, shrubs, etc.) are thus needed to further advance our understanding of this issue” (Page 12, line 256-258).

Overall, we appreciate for the reviewer’s insightful comments. These comments

enabled us to have a comprehensive overview on published literatures in priming area, and a deeper thinking on the experimental method, error estimation, results interpretation and future directions, and guided us to have a thorough revision on the MS. By addressing these comments, we feel that the revised MS has been greatly improved. Thank you!

Response to Reviewer #3:

[Comment 1] The manuscript by Chen et al is an interesting attempt to estimate the importance of different carbon stabilization process (chemical recalcitrance and mineral interactions) on priming intensity. Priming is considered as an important process controlling soil carbon fluxes but we still lack the understanding of the mechanisms behind. Consequently, predicting the priming intensity is very challenging. I appreciate the efforts made by the authors to sample over a large region (the Tibetan plateau) but I have several concerns mainly about methodological aspects that should be considered before any publications.

[Response] Thanks for the reviewer's positive comments.

[Comment 2] 1. A large part of the manuscript relate priming and chemical recalcitrance but at any moment, the important debate occurring in the soil science about chemical recalcitrance (Dungait et al., 2012; Kleber, 2010; Marschner et al., 2008; Mikutta et al., 2006) is mentioned. Even though, the concept is not totally abandoned by the soil science community it is largely criticized (Lehmann and Kleber, 2015; Schmidt et al., 2011) and more and more evidences tends to suggest that this concept is misleading (e.g. Klotzbücher et al., 2011). The authors may disagree and think the concept is still useful but they must explain why and mention how their arguments take part of the current debate.

[Response] Very good comment! We acknowledge that the perceived importance of chemical recalcitrance in governing the turnover of SOM have been increasingly challenged by recent work (Kleber, 2010, Schmidt *et al.*, 2011, Lehmann & Kleber, 2015). Nevertheless, **despite of these controversy, we think the chemical stabilized mechanism is still useful and should not be totally abandoned due to the fact that** the impacts of SOM composition on soil C dynamics depend on the status of mineral protection (Mikutta *et al.*, 2006, Dungait *et al.*, 2012, Sjögersten *et al.*, 2016). When soils have higher clay mineral content, the SOM stability is governed by mineral-organic association and aggregate occlusion rather than the chemical recalcitrance (Mikutta *et al.*, 2006, Marschner *et al.*, 2008, Dungait *et al.*, 2012).

Nevertheless, the chemical recalcitrance could still regulate SOM decomposition when SOM is weakly stabilized by physico-chemical interactions (Sjögersten *et al.*, 2016). This is **the case for dissolved organic C (DOC), free light-fraction SOC and to a certain extent to OC in soil with less mineral-organic associations** (Marschner *et al.*, 2008). In line with this argument, the molecular structure and functional group of **DOC** are generally considered as the main factor that controls its biodegradability (Marschner & Kalbitz, 2003, Woods *et al.*, 2011, Drake *et al.*, 2015). Similarly, **in mineral-free SOM pool**, the abundance of aromatic plus alkyl-C relative to O-alkyl-C groups was found to drive the Q_{10} of SOM decomposition (Wagai *et al.*, 2013). Moreover, the regulation of CO₂ release by chemical composition of SOM was also frequently observed **in permafrost soils** (Hodgkins *et al.*, 2014, Treat *et al.*, 2014, Sjögersten *et al.*, 2016), where the mineral-organic associations have been regarded as less important than in other soils (Hofle *et al.*, 2013, Ping *et al.*, 2015). Given that only 15.8% of SOC pool was protected by minerals in our study area (Fang *et al.*, 2019), SOM decomposition could then be jointly affected by its chemical composition and mineral protection. **That's also why we incorporated the chemical recalcitrance and physico-chemical protection simultaneously into the term 'SOM stability' in current study.** We have added one paragraph to clearly mention these points in the *Discussion* section of the revised MS (Page 11-12, line 228-241). Nevertheless, we agree that the term of 'chemical recalcitrance' maybe not appropriate (Kleber, 2010), and we are happy to change it. Could you please suggest one?

*[Comment 3] 2. To measure priming, the authors scaled the amount of glucose on the microbial biomass but they do all their analyses with priming given in gC-CO₂ g⁻¹ of soil without considering that priming intensity is related to the labile material amended (e.g. Xiao *et al.*, 2015). I suggest to redo the analysis estimating priming per g of C soil and g C-glucose added to be consistent*

[Response] Very good comment! Following the reviewer's comment, we have also presented the priming effect in other two units and re-analysed the data. The

results of re-analysis also showed that SOM stability explained much more variance in relative priming effect than plant, soil and microbial properties, which was in consistent with our conclusion (Fig. R17).

Despite of the consistency, we prefer to only use per unit soil weight and % rather than per unit SOC and per unit C added as unit of priming effect due to the following reason: The purpose of our study was not to compare the magnitude of priming effect among sites, but rather to quantify the relative importance of different factors (*i.e.*, plant, soil, and microbial properties, and SOM stability) in regulating the priming effect over the broad geographic scale. **To achieve this aim, we need to include as many potential drivers as possible in the multivariate analysis.** The SOC content (Kuzyakov, 2000) and the amount of input C (Xiao *et al.*, 2015) are exactly two important variables that have been proved to regulate the priming effects. Thus, we have included these two variables as examined factor in soil (e.g., SOC, TN, pH...) and plant (e.g., ANPP, **the amount of input C...**) properties. Nevertheless, **if we used per unit SOC and per unit C added as the unit of priming effect, we could not evaluate their relationship with priming effect due to the autocorrelation and could not further compare their relative importance with other predictors either.** Probably, **this is also the reason why the priming effect was presented in per unit soil dry weight in the recent two large-scale empirical studies** (Razanamalala *et al.*, 2017, Perveen *et al.*, 2019). Due to this point, we prefer to only use per unit soil weight and percentage of stimulation of flux as unit of priming effect in the revised MS. Nevertheless, we can add the results using per unit SOM or C added if the reviewer persists his/her opinion. Thanks for your understanding!

Figure R17. Relative contribution of SOM stability, and plant, soil and microbial properties in driving priming effect represented in per unit soil weight (a), per unit soil organic carbon (b) and per unit added C (c).

[Comment 4] 3. The modelling approach is difficult to understand and it took me several reading to finally get something but I am even not sure that I understood completely. Basically, the authors considered that with such short term incubations it is not possible to estimate the most passive pool and they therefore fixed the C_{tot} parameters of eq. 5 to be 1000mg g^{-1} . Based on this assumptions they estimate the f_l parameter and the two decay rates of the corresponding pools. C_{tot} is therefore not the total soil C stocks but the sum of the fast and slow pools. Therefore, the fig. S2b is misleading because it suggests that you correlated the priming with the fast pool size whereas it is actually just the fast pool over the sum of the fast and slow pools. Moreover, much more information are needed in particular the prior values, the estimated parameters and their confidence interval.

[Response] Sorry for the confusion. As pointed out by the reviewer, C_{tot} is not the

absolute amount of soil C stock, but the maximum of C loss percentage, which equals to 100 %. Given that the unit of CO₂ emission rate is expressed as per unit C (*i.e.*, mg C g⁻¹ C d⁻¹), C_{tot} is fixed to 1000 mg C g⁻¹ C (equals to 100%) (Chen *et al.*, 2016b). Additionally, as mentioned by the reviewer, f_1 is the proportion of fast pool, which is the fast pool over the sum of the fast and slow pools. To avoid the confusion, we have changed the term of ‘fast pool size’ to ‘the proportion of fast C pool’ in the Method session and Fig. S3 of revised MS (Page 22-23, line 463-468).

Following the reviewer’s comment, we have also added more information on the prior values, the estimated parameters and their confidence intervals as follows:

“where $R(t)$ is the CO₂ emission rate (mg C g⁻¹ SOC d⁻¹) at time t , C_{tot} is the maximum C loss percentage (that is, 100% = 1000 mg C g⁻¹ SOC), f_1 is the proportion of the fast pool, and k_1 and k_2 are the decay rates of the fast and slow pools (day⁻¹), respectively. The three parameters (f_1 , k_1 and k_2) were estimated by the Markov Chain Monte Carlo (MCMC) approach (Liang *et al.*, 2015). Before applying the MCMC approach, the prior parameter range in the initial model (Table R5) was set as wide as possible to cover the possibilities for all study sites (Schädel *et al.*, 2014). After the MCMC simulations, maximum likelihood estimates were used to quantify the well-constrained parameters (Chen *et al.*, 2016a), while mean values were calculated for the poorly-constrained parameters (Table R6).”

Table R5. Prior parameter ranges for C pool partitioning coefficients (f_i) and decay rates (k_i) parameter.

Parameter	Description	Lower limit	Upper limit
f_1	proportion of fast C pool	0	0.1
k_1	decay rate of fast C pool	0	1
k_2	decay rate of slow C pool	0	0.01

C pool proportion is unit less, decay rates are day⁻¹

Parameter ranges were estimated according to previous studies (Schädel *et al.*, 2013, Schädel *et al.*, 2014).

Table R6. Maximum likelihood estimates (MLEs) of posterior probability density functions of model parameters for the 30 study sites.

Site No.	Parameter		
	$k_1 (\times 10^{-2})$	$k_2 (\times 10^{-5})$	$f_1 (\times 10^{-2})$
1	3.50 [3.48, 3.52]	1.25 [1.21, 1.29]	1.14 [1.13, 1.14]
2	2.46 [2.44, 2.48]	2.05 [1.80, 2.30]	1.06 [1.05, 1.06]
3	10.7 [10.3, 11.1]	7.21 [6.99, 7.44]	0.60 [0.58, 0.61]
4	5.03 [4.95, 5.10]	1.53 [1.48, 1.59]	0.47 [0.47, 0.48]
5	4.58 [4.53, 4.62]	1.24 [1.20, 1.29]	0.49 [0.49, 0.50]
6	36.5 [36.4, 36.6]	7.31 [7.27, 7.35]	0.07 [0.07, 0.08]
7	3.88 [3.85, 3.90]	3.17 [3.03, 3.30]	2.90 [2.88, 2.92]
8	4.38 [4.34, 4.41]	0.60 [0.58, 0.62]	0.47 [0.46, 0.47]
9	3.56 [3.52, 3.59]	0.63 [0.60, 0.67]	0.63 [0.62, 0.63]
10	3.62 [3.60, 3.66]	2.70 [2.63, 2.77]	1.11 [1.10, 1.12]
11	4.55 [4.53, 4.57]	1.37 [1.35, 1.38]	0.73 [0.73, 0.74]
12	2.91 [2.87, 2.94]	3.70 [3.49, 3.90]	2.10 [2.08, 2.13]
13	3.84 [3.81, 3.87]	2.69 [2.61, 2.76]	1.61 [1.60, 1.62]
14	3.79 [3.76, 3.81]	10.9 [10.5, 11.2]	4.84 [4.81, 4.88]
15	3.29 [3.26, 3.31]	1.71 [1.65, 1.77]	1.18 [1.18, 1.19]
16	3.59 [3.55, 3.64]	4.10 [3.96, 4.24]	1.48 [1.47, 1.50]
17	34.4 [32.8, 36.0]	59.0 [57.7, 60.3]	1.42 [1.40, 1.44]
18	79.9 [79.3, 80.6]	83.4 [83.1, 83.8]	0.45 [0.45, 0.46]
19	73.6 [72.7, 74.5]	72.0 [71.6, 72.4]	0.45 [0.44, 0.46]
20	4.06 [4.04, 4.07]	2.78 [2.67, 2.88]	4.49 [4.47, 4.50]
21	2.19 [2.14, 2.23]	22.9 [22.7, 23.2]	1.48 [1.45, 1.51]
22	24.1 [23.1, 25.1]	42.6 [42.3, 42.8]	0.44 [0.43, 0.46]
23	42.2 [40.3, 44.1]	93.8 [93.2, 94.5]	1.86 [1.76, 1.97]
24	47.0 [46.1, 47.9]	92.3 [92.0, 92.7]	0.54 [0.53, 0.55]
25	3.59 [3.57, 3.61]	18.5 [18.0, 19.1]	8.15 [8.11, 8.19]
26	2.36 [2.35, 2.37]	37.9 [37.8, 38.0]	9.72 [9.71, 9.74]
27	3.90 [3.87, 3.93]	6.08 [5.87, 6.30]	3.82 [3.79, 3.85]
28	13.8 [13.1, 14.5]	523.6 [520.7, 526.5]	2.57 [2.49, 2.64]
29	42.1 [40.5, 43.8]	32.2 [32.0, 32.4]	0.18 [0.17, 0.19]
30	8.64 [8.43, 8.85]	23.9 [23.7, 24.0]	0.41 [0.40, 0.42]

k_1 and k_2 are the decay rates of fast and slow pools, respectively. f_1 is the proportion of the fast pool. The interquartile range is presented in square brackets.

[Comment 5] Minor comments: L49: Don't forget lateral transfer through erosion for instance

[Response] Thanks for the reviewer's reminder. Following the reviewer's comment, we have rephrased the sentence as follows: "As the largest C pool in terrestrial ecosystems, the soil C pool size is determined by the balance between C inputs from plant production and C outputs through microbial decomposition (Jackson *et al.*, 2017) and lateral transfer into aquatic systems (Battin *et al.*, 2009)"(Page 3, line 52-54).

[Comment 6] How do you stored the samples before incubations?

[Response] Due to the harsh environment (*i.e.*, high altitude, low oxygen, and sparse roadwork), it is difficult to conduct large-scale survey on the Tibetan Plateau. Thus, it costs us two years to collect these plant and soil samples across 30 sites along an approximately 2200 km grassland transect in this unique geographic region. To ensure the consistent incubation condition for all the study sites, as done in many literatures (Kane *et al.*, 2013, Roy Chowdhury *et al.*, 2014, Treat *et al.*, 2014), soil samples were stored in the freezer before incubations. To reduce the possible disturbance from manipulation (*i.e.*, freeze-thaw) incubation experiments, soil samples were pre-incubated for 7 days at 15 °C before measuring soil CO₂ release. Thanks for your understanding!

[Comment 7] L285: How do you estimate the microbial biomass? The amount of glucose added as well as the microbial biomass should be given in sup mat

[Response] The MBC was analyzed using the chloroform fumigation method with the conversion factors of 0.45 (Joergensen, 1996). We have added this information in the *Methods* section of the revised MS (Page 19, line 397-399), and also added the microbial biomass data which equals to the amount of glucose added in the supplementary Data 1.

[Comment 8] L286: I don't get why you had blank if you started with CO2-free

atmosphere. The incubation flasks were not closed?

[Response] Yes, the bottles were closed during the incubation. The blanks were set to eliminate background $\delta^{13}\text{C}$ noise (Subke *et al.*, 2004, De Troyer *et al.*, 2011, Guelland *et al.*, 2013). This is because we were not sure whether the effect of remaining CO_2 concentration on the C isotope can be ignored, although bottles were flushed with CO_2 -free atmosphere for 20 minutes before each time's measurement (Fontaine *et al.*, 2004, Wang *et al.*, 2015).

[Comment 9] L320: *I am ok with this assumption because your sampling points are mainly on grassland and steppe which, I guess, are largely composed of annual plants but it should be a bit more explained in the text.*

[Response] Actually, the dominant species across the alpine grassland are perennial herbs, such as *Stipa purpurea*, *Carex moorcroftii* and *Kobresia pygmaea*. However, the aboveground parts of these perennials still die in non-growing season (Photo R1). Therefore, the peak aboveground biomass in the growing season could be considered as the aboveground net primary production (ANPP) (Scurlock *et al.*, 2002). We have clearly mentioned these points in the *Method* session of the revised MS (Page 15, line 302-304). To avoid the confusion, we have also revised aboveground biomass to ANPP in the revised MS (Page 19, line 390).

Photo R1. Landscape of a typical alpine grassland during non-growing season.

[Comment 10] L342: *Please rephrase, I guess your talking about the Bosso et al protocol but with the “Nat. Com.” style the reading is not straightforward.*

[Response] Done as suggested.

Overall, we appreciate for the reviewer's insightful comments. These comments enabled us to have a deeper thinking on the related concept, data analysis and modeling details and guided us to connect our work with existing theories and empirical results. By addressing these comments, we feel that the revised MS has been greatly improved. Thank you!

References:

- Bastida F, Torres IF, Hernández T, Bombach P, Richnow HH, García C (2013) Can the labile carbon contribute to carbon immobilization in semiarid soils? Priming effects and microbial community dynamics. *Soil Biology & Biochemistry*, **57**, 892-902.
- Blagodatskaya E, Kuzyakov Y (2008) Mechanisms of real and apparent priming effects and their dependence on soil microbial biomass and community structure: critical review. *Biology and Fertility of Soils*, **45**, 115-131.
- Blagodatskaya EV, Blagodatsky SA, Anderson TH, Kuzyakov Y (2007) Priming effects in Chernozem induced by glucose and N in relation to microbial growth strategies. *Applied Soil Ecology*, **37**, 95-105.
- Bradford MA, Wieder WR, Bonan GB, Fierer N, Raymond PA, Crowther TW (2016) Managing uncertainty in soil carbon feedbacks to climate change. *Nature Climate Change*, **6**, 751-758.
- Chapin FS, Matson PA, Vitousek PM (2012) Principles of Terrestrial Ecosystem Ecology, 2nd edn. Springer, New York.
- Chen L, Liang J, Qin S *et al.* (2016a) Determinants of carbon release from the active layer and permafrost deposits on the Tibetan Plateau. *Nature Communications*, **7**, 13046.
- Chen LY, Liang JY, Qin SQ *et al.* (2016b) Determinants of carbon release from the active layer and permafrost deposits on the Tibetan Plateau. *Nature Communications*, **7**, 13046.
- Chen R, Senbayram M, Blagodatsky S *et al.* (2014) Soil C and N availability determine the priming effect: microbial N mining and stoichiometric decomposition theories. *Global Change Biology*, **20**, 2356-2367.
- Chen YL, Chen LY, Peng YF *et al.* (2016c) Linking microbial C:N:P stoichiometry to microbial community and abiotic factors along a 3500-km grassland transect on the Tibetan Plateau. *Global Ecology and Biogeography*, **25**, 1416-1427.
- Chen YL, Ding JZ, Peng YF *et al.* (2016d) Patterns and drivers of soil microbial communities in Tibetan alpine and global terrestrial ecosystems. *Journal of Biogeography*, **43**, 2027-2039.
- Cheng WX, Parton WJ, Gonzalez-Meler MA *et al.* (2014) Synthesis and modeling perspectives of rhizosphere priming. *New Phytologist*, **201**, 31-44.
- Chowdhury S, Farrell M, Bolan N (2014) Priming of soil organic carbon by malic acid addition is differentially affected by nutrient availability. *Soil Biology & Biochemistry*, **77**, 158-169.
- Colman BP, Schimel JP (2013) Drivers of microbial respiration and net N mineralization at the continental scale. *Soil Biology & Biochemistry*, **60**, 65-76.
- Craine JM, Fierer N, Mclauchlan KK (2010) Widespread coupling between the rate and temperature sensitivity of organic matter decay. *Nature Geoscience*, **3**, 854-857.
- De Baets S, Van De Weg MJ, Lewis R *et al.* (2016) Investigating the controls on soil organic matter decomposition in tussock tundra soil and permafrost after fire.

- Soil Biology & Biochemistry*, **99**, 108-116.
- De Troyer I, Amery F, Van Moorleghem C, Smolders E, Merckx R (2011) Tracing the source and fate of dissolved organic matter in soil after incorporation of a ¹³C labelled residue: A batch incubation study. *Soil Biology & Biochemistry*, **43**, 513-519.
- Ding JZ, Chen LY, Zhang BB *et al.* (2016a) Linking temperature sensitivity of soil CO₂ release to substrate, environmental, and microbial properties across alpine ecosystems. *Global Biogeochemical Cycles*, **30**, 1310-1323.
- Ding JZ, Li F, Yang GB *et al.* (2016b) The permafrost carbon inventory on the Tibetan Plateau: a new evaluation using deep sediment cores. *Global Change Biology*, **22**, 2688-2701.
- Doetterl S, Stevens A, Six J *et al.* (2015) Soil carbon storage controlled by interactions between geochemistry and climate. *Nature Geoscience*, **8**, 780-783.
- Drake TW, Wickland KP, Spencer RGM, Mcknight DM, Striegl RG (2015) Ancient low-molecular-weight organic acids in permafrost fuel rapid carbon dioxide production upon thaw. *Proceedings of the National Academy of Sciences of the United States of America*, **112**, 13946-13951.
- Dungait JaJ, Hopkins DW, Gregory AS, Whitmore AP (2012) Soil organic matter turnover is governed by accessibility not recalcitrance. *Global Change Biology*, **18**, 1781-1796.
- Fang K, Qin S, Chen L, Zhang Q, Yang Y (2019) Al/Fe mineral controls on soil organic carbon stock across Tibetan alpine grasslands. *Journal of Geophysical Research: Biogeosciences*, 10.1029/2018JG004782.
- Fierer N, Colman BP, Schimel JP, Jackson RB (2006) Predicting the temperature dependence of microbial respiration in soil: A continental-scale analysis. *Global Biogeochemical Cycles*, **20**. doi:10.1029/2005GB002644.
- Finley BK, Dijkstra P, Rasmussen C *et al.* (2018) Soil mineral assemblage and substrate quality effects on microbial priming. *Geoderma*, **322**, 38-47.
- Fontaine S, Bardoux G, Abbadie L, Mariotti A (2004) Carbon input to soil may decrease soil carbon content. *Ecology Letters*, **7**, 314-320.
- Fontaine S, Mariotti A, Abbadie L (2003) The priming effect of organic matter: a question of microbial competition? *Soil Biology & Biochemistry*, **35**, 837-843.
- Gentsch N, Wild B, Mikutta R *et al.* (2018) Temperature response of permafrost soil carbon is attenuated by mineral protection. *Global Change Biology*, doi: 10.1111/gcb.14316.
- Guelland K, Esperschuetz J, Bornhauser D, Bernasconi SM, Kretzschmar R, Hagedorn F (2013) Mineralisation and leaching of C from C-13 labelled plant litter along an initial soil chronosequence of a glacier forefield. *Soil Biology & Biochemistry*, **57**, 237-247.
- Guenet B, Camino-Serrano M, Ciais P, Tifafi M, Maignan F, Soong JL, Janssens IA (2018) Impact of priming on global soil carbon stocks. *Global Change Biology*, **24**, 1873-1883.
- Hamer U, Marschner B (2005) Priming effects in soils after combined and repeated

- substrate additions. *Geoderma*, **128**, 38-51.
- Hobbie JE, Hobbie EA (2013) Microbes in nature are limited by carbon and energy: the starving-survival lifestyle in soil and consequences for estimating microbial rates. *Front Microbiology*, **4**, 324.
- Hodgkins SB, Tfaily MM, Mccalley CK *et al.* (2014) Changes in peat chemistry associated with permafrost thaw increase greenhouse gas production. *Proceedings of the National Academy of Sciences of the United States of America*, **111**, 5819-5824.
- Hofle S, Rethemeyer J, Mueller CW, John S (2013) Organic matter composition and stabilization in a polygonal tundra soil of the Lena Delta. *Biogeosciences*, **10**, 3145-3158.
- Hopkins FM, Filley TR, Gleixner G, Lange M, Top SM, Trumbore SE (2014) Increased belowground carbon inputs and warming promote loss of soil organic carbon through complementary microbial responses. *Soil Biology & Biochemistry*, **76**, 57-69.
- Howard DM, Howard PJA (1993) Relationships between CO₂ evolution, moisture-content and temperature for a range of soil types. *Soil Biology & Biochemistry*, **25**, 1537-1546.
- Huo C, Luo Y, Cheng W (2017) Rhizosphere priming effect: A meta-analysis. *Soil Biology & Biochemistry*, **111**, 78-84.
- Jackson RB, Lajtha K, Crow SE, Hugelius G, Kramer MG, Piñeiro G (2017) The Ecology of Soil Carbon: Pools, Vulnerabilities, and Biotic and Abiotic Controls. *Annual Review of Ecology Evolution and Systematics*, **48**, 419-445.
- Joergensen RG (1996) The fumigation-extraction method to estimate soil microbial biomass: Calibration of the k_{EC} value. *Soil Biology & Biochemistry*, **28**, 25-31.
- Kane ES, Chivers MR, Turetsky MR *et al.* (2013) Response of anaerobic carbon cycling to water table manipulation in an Alaskan rich fen. *Soil Biology & Biochemistry*, **58**, 50-60.
- Kleber M (2010) What is recalcitrant soil organic matter? *Environmental Chemistry*, **7**, 320-332.
- Kuzyakov Y, Bol R (2006) Sources and mechanisms of priming effect induced in two grassland soils amended with slurry and sugar. *Soil Biology & Biochemistry*, **38**, 747-758.
- Kuzyakov Y, Friedel J.K., Stahr, K. (2000) Review of mechanisms and quantification of priming effects. *Soil Biology & Biochemistry*, **32**, 1485-1498.
- Lehmann J, Kleber M (2015) The contentious nature of soil organic matter. *Nature*, **528**, 60-68.
- Liang J, Li D, Shi Z *et al.* (2015) Methods for estimating temperature sensitivity of soil organic matter based on incubation data: A comparative evaluation. *Soil Biology & Biochemistry*, **80**, 127-135.
- Luo Z, Feng W, Luo Y, Baldock J, Wang E (2017) Soil organic carbon dynamics jointly controlled by climate, carbon inputs, soil properties and soil carbon fractions. *Global Change Biology*, **23**, 4430-4439.
- Luo Z, Wang E, Smith C (2015) Fresh carbon input differentially impacts soil carbon

- decomposition across natural and managed systems. *Ecology*, **96**, 2806-2813.
- Luo Z, Wang E, Sun OJ (2016) A meta-analysis of the temporal dynamics of priming soil carbon decomposition by fresh carbon inputs across ecosystems. *Soil Biology & Biochemistry*, **101**, 96-103.
- Marschner B, Brodowski S, Dreves A *et al.* (2008) How relevant is recalcitrance for the stabilization of organic matter in soils? *Journal of Plant Nutrition and Soil Science*, **171**, 91-110.
- Marschner B, Kalbitz K (2003) Controls of bioavailability and biodegradability of dissolved organic matter in soils. *Geoderma*, **113**, 211-235.
- Mau RL, Liu CM, Aziz M *et al.* (2015) Linking soil bacterial biodiversity and soil carbon stability. *ISME Journal*, **9**, 1477-1480.
- Mikutta R, Kleber M, Torn MS, Jahn R (2006) Stabilization of soil organic matter: association with minerals or chemical recalcitrance? *Biogeochemistry*, **77**, 25-56.
- Mooshammer M, Hofhansl F, Frank AH *et al.* (2017) Decoupling of microbial carbon, nitrogen, and phosphorus cycling in response to extreme temperature events. *Science Advances*, **3**, e1602781.
- Peng S, Ciais P, Chevallier F *et al.* (2015) Benchmarking the seasonal cycle of CO₂ fluxes simulated by terrestrial ecosystem models. *Global Biogeochemical Cycles*, **29**, 46-64.
- Perveen N, Barot S, Alvarez G *et al.* (2014) Priming effect and microbial diversity in ecosystem functioning and response to global change: a modeling approach using the SYMPHONY model. *Global Change Biology*, **20**, 1174-1190.
- Perveen N, Barot S, Maire V *et al.* (2019) Universality of priming effect: An analysis using thirty five soils with contrasted properties sampled from five continents. *Soil Biology & Biochemistry*, **134**, 162-171.
- Phillips RP, Ehlitz Y, Bier R, Bernhardt ES (2008) New approach for capturing soluble root exudates in forest soils. *Functional Ecology*, **22**, 990-999.
- Ping CL, Jastrow JD, Jorgenson MT, Michaelson GJ, Shur YL (2015) Permafrost soils and carbon cycling. *Soil*, **1**, 147-171.
- Qiao N, Schaefer D, Blagodatskaya E, Zou X, Xu X, Kuzyakov Y (2014) Labile carbon retention compensates for CO₂ released by priming in forest soils. *Global Change Biology*, **20**, 1943-1954.
- Qiao N, Xu X, Hu Y, Blagodatskaya E, Liu Y, Schaefer D, Kuzyakov Y (2016) Carbon and nitrogen additions induce distinct priming effects along an organic-matter decay continuum. *Scientific Reports*, **6**, 19865.
- Razanamalala K, Razafimbelo T, Maron P-A *et al.* (2017) Soil microbial diversity drives the priming effect along climate gradients: a case study in Madagascar. *The ISME Journal*, **12**, 451.
- Reinsch S, Ambus P, Thornton B, Paterson E (2013) Impact of future climatic conditions on the potential for soil organic matter priming. *Soil Biology & Biochemistry*, **65**, 133-140.
- Rey ANA, Jarvis P (2006) Modelling the effect of temperature on carbon mineralization rates across a network of European forest sites (FORCAST).

- Global Change Biology*, **12**, 1894-1908.
- Rowley MC, Grand S, Verrecchia ÉP (2018) Calcium-mediated stabilisation of soil organic carbon. *Biogeochemistry*, **137**, 27-49.
- Roy Chowdhury T, Herndon EM, Phelps TJ *et al.* (2014) Stoichiometry and temperature sensitivity of methanogenesis and CO₂ production from saturated polygonal tundra in Barrow, Alaska. *Global Change Biology*, 722-737.
- Schädel C, Luo YQ, Evans RD, Fei SF, Schaeffer SM (2013) Separating soil CO₂ efflux into C-pool-specific decay rates via inverse analysis of soil incubation data. *Oecologia*, **171**, 721-732.
- Schädel C, Schuur EaG, Bracho R *et al.* (2014) Circumpolar assessment of permafrost C quality and its vulnerability over time using long-term incubation data. *Global Change Biology*, **20**, 641-652.
- Schmidt MW, Torn MS, Abiven S *et al.* (2011) Persistence of soil organic matter as an ecosystem property. *Nature*, **478**, 49-56.
- Scurlock JMO, Johnson K, Olson RJ (2002) Estimating net primary productivity from grassland biomass dynamics measurements. *Global Change Biology*, **8**, 736-753.
- Shi XZ, Yu DS, Warner ED, Pan XZ, Petersen GW, Gong ZG, Weindorf DC (2004) Soil Database of 1:1,000,000 Digital Soil Survey and Reference System of the Chinese Genetic Soil Classification System. **45**.
- Sjögersten S, Caul S, Daniell TJ, Jurd APS, O'sullivan OS, Stapleton CS, Titman JJ (2016) Organic matter chemistry controls greenhouse gas emissions from permafrost peatlands. *Soil Biology & Biochemistry*, **98**, 42-53.
- Song M, Guo Y, Yu F, Zhang X, Cao G, Cornelissen JHC (2018) Shifts in priming partly explain impacts of long-term nitrogen input in different chemical forms on soil organic carbon storage. *Global Change Biology*. DOI: 10.1111/gcb.14304
- Subke JA, Hahn V, Battipaglia G, Linder S, Buchmann N, Cotrufo MF (2004) Feedback interactions between needle litter decomposition and rhizosphere activity. *Oecologia*, **139**, 551-559.
- Sullivan BW, Hart SC (2013) Evaluation of mechanisms controlling the priming of soil carbon along a substrate age gradient. *Soil Biology & Biochemistry*, **58**, 293-301.
- Sulman BN, Phillips RP, Oishi AC, Shevliakova E, Pacala SW (2014) Microbe-driven turnover offsets mineral-mediated storage of soil carbon under elevated CO₂. *Nature Climate Change*, **4**, 1099-1102.
- Thiessen S, Gleixner G, Wutzler T, Reichstein M (2013) Both priming and temperature sensitivity of soil organic matter decomposition depend on microbial biomass – An incubation study. *Soil Biology & Biochemistry*, **57**, 739-748.
- Tian H, Lu C, Yang J *et al.* (2015) Global patterns and controls of soil organic carbon dynamics as simulated by multiple terrestrial biosphere models: Current status and future directions. *Global Biogeochemical Cycles*, **29**, 2014GB005021.
- Treat CC, Wollheim WM, Varner RK, Grandy AS, Talbot J, Froelking S (2014)

- Temperature and peat type control CO₂ and CH₄ production in Alaskan permafrost peats. *Global Change Biology*, **20**, 2674-2686.
- Wagai R, Kishimoto-Mo AW, Yonemura S, Shirato Y, Hiradate S, Yagasaki Y (2013) Linking temperature sensitivity of soil organic matter decomposition to its molecular structure, accessibility, and microbial physiology. *Global Change Biology*, **19**, 1114-1125.
- Wang H, Xu W, Hu G, Dai W, Jiang P, Bai E (2015) The priming effect of soluble carbon inputs in organic and mineral soils from a temperate forest. *Oecologia*, **178**, 1239-1250.
- Wild B, Schneck J, Alves RJE *et al.* (2014) Input of easily available organic C and N stimulates microbial decomposition of soil organic matter in arctic permafrost soil. *Soil Biology & Biochemistry*, **75**, 143-151.
- Woods GC, Simpson MJ, Pautler BG, Lamoureux SF, Lafrenière MJ, Simpson AJ (2011) Evidence for the enhanced lability of dissolved organic matter following permafrost slope disturbance in the Canadian High Arctic. *Geochimica et Cosmochimica Acta*, **75**, 7226-7241.
- Xu T, White L, Hui DF, Luo YQ (2006) Probabilistic inversion of a terrestrial ecosystem model: Analysis of uncertainty in parameter estimation and model prediction. *Global Biogeochemical Cycles*, **20**, GB2007.
- Yang YH, Fang JY, Pan YD, Ji CJ (2009) Aboveground biomass in Tibetan grasslands. *Journal of Arid Environments*, **73**, 91-95.
- Ye RZ, Doane TA, Morris J, Horwath WR (2015) The effect of rice straw on the priming of soil organic matter and methane production in peat soils. *Soil Biology & Biochemistry*, **81**, 98-107.
- Zhu B, Cheng W (2011) Constant and diurnally-varying temperature regimes lead to different temperature sensitivities of soil organic carbon decomposition. *Soil Biology & Biochemistry*, **43**, 866-869.

Nature Research Editing Service Certification

This is to certify that the manuscript titled Regulation of priming effect by soil organic matter stability over broad geographic scale was edited for English language usage, grammar, spelling and punctuation by one or more native English-speaking editors at Nature Research Editing Service. The editors focused on correcting improper language and rephrasing awkward sentences, using their scientific training to point out passages that were confusing or vague. Every effort has been made to ensure that neither the research content nor the authors' intentions were altered in any way during the editing process.

Documents receiving this certification should be English-ready for publication; however, please note that the author has the ability to accept or reject our suggestions and changes. To verify the final edited version, please visit our verification page. If you have any questions or concerns over this edited document, please contact Nature Research Editing Service at support@as.springernature.com.

Manuscript title: Regulation of priming effect by soil organic matter stability over broad geographic scale

Authors: Leiyi Chen, Li Liu, Shuqi Qin, Guibiao Yang, Kai Fang, Biao Zhu, Yakov Kuzyakov, Yunping Xu, Yuanhe Yang

Key: 1F85-AF53-A030-ED58-A2EP

This certificate may be verified at secure.authorservices.springernature.com/certificate/verify.

Nature Research Editing Service is a service from Springer Nature, one of the world's leading research, educational and professional publishers. We have been a reliable provider of high-quality editing since 2008.

Nature Research Editing Service comprises a network of more than 900 language editors with a range of academic backgrounds. All our language editors are native English speakers and must meet strict selection criteria. We require that each editor has completed or is completing a Masters, Ph.D. or M.D. qualification, is affiliated with a top US university or research institute, and has undergone substantial editing training. To ensure we can meet the needs of researchers in a broad range of fields, we continually recruit editors to represent growing and new disciplines.

Uploaded manuscripts are reviewed by an editor with a relevant academic background. Our senior editors also quality-assess each edited manuscript before it is returned to the author to ensure that our high standards are maintained.

Reviewers' comments:

Reviewer #1 (Remarks to the Author):

Main comments:

Chen et al. have responded to all my previous comments, and those of the other reviewers, and I think the analysis in the paper has improved substantially. In general, the inclusion of the relative priming effects analysis has improved the confidence in the findings (but see comment below in relation to the partial correlations). By responding to reviewer 2 and including more plant variables, it appears that the main conclusion should now be that organic matter quality, as controlled by plant community composition and rates of input, determines the potential for glucose addition to induce priming effects. The slightly artificial distinction between plant variables and SOM quality is an issue that the authors address in their response to reviewer 2, when they consider how the labile C pool is controlled by the plant community. However, the fact that SOM quality is not truly independent of the plant community should be discussed more in the manuscript given the greater role plant variables play, both directly and indirectly, in the new analysis.

It is the analysis presented in Figure 2 (and Supplementary figure 4) that I think is most important. In the new version of this analysis, forbs and sedges seem to retain or increase their importance even after the direct measures of SOM stability are accounted for. Furthermore, for the SOM stability variables, when C input is included, the importance of all the SOM stability variables declines substantially. I don't think this undermines the overall conclusion of the paper but I think the authors do need to explain more clearly how and why they have separated the different variables into the different groups, and to discuss more how many of the SOM stability variables (especially the labile pool size and the chemical assays) are controlled by the quality and quantity of the inputs (perhaps as suggested by the structural equation modelling).

The authors now present most of the analyses for both relative and absolute priming effects, but I think it is important that the authors also show the analysis in Figure 2 for both absolute and the relative priming effects. I apologise if I have missed this but I couldn't find the partial correlations analysis for relative priming effects. I'm also unclear why root biomass, which has been added to the plant properties does not seem to appear in Figure 2 but does appear in Supplementary Figure 4.

Overall, the analysis appears more robust and the dataset is very impressive. The SOM stability variables still appear to be the most important determinants of the glucose-induced priming effects,

but I think the role of plant community composition and C input rate in determining these SOM stability variables needs to be emphasised more strongly.

Minor comments:

Line 5: I think it needs to be clearer what the soil properties are that are not related to chemical recalcitrance or physico-chemical protection. This is not really made explicitly clear until the methods section on line 510. The distinction needs to be made earlier.

Line 187 to 189: I recognise that a proof reading service has been used but there are still a few poorly phrased sentences and this sentence contains some typos. Should it be 'indirect effects'?

Line 207:208: It would be helpful to explain why high SOM recalcitrance would trigger greater microbial N demand.

Lines 234-241: Despite the growing recognition of the role physico-chemical protection plays in controlling soil C storage in soils from all regions, I think it is probably still fair to suggest that C-rich soils in cool regions may have greater amounts of unprotected C than soils in other regions. Therefore, I wonder whether the authors need to consider the extent to which the findings would apply to soils in warmer climates.

Reviewer #2 (Remarks to the Author):

The reviewers have done considerable work in response to my comments and to those of the other two reviewers. Given their effort, I think the paper should be acceptable in Nature Communications. Found below are a few final suggestions for improvement.

The one item where I think more description is still needed is in the selection of sites (see next point, as well). The paper states that the sites were selected for purposes of the gradient. How so, though, and for which gradients? All of the variables equally (see immediately below)? I'm only asking for more specifics about the process for site selection.

I think the manuscript would also be easier to grasp if all of the information about the gradient wasn't in a Supplementary table: "These sites covered a wide range of climate (Supplementary Table 1), vegetation productivity, soil and microbial properties (see details in the methods, Supplementary Data 1). Can't the authors include even a brief description of the gradients in this paragraph (lines 105-107)? This sentence from the Supplement, for instance, would be an improvement (lines 293-299): "These sites were set throughout the geographical extent of alpine grasslands, covering a wide range of climates (mean annual temperature: -3.7–6.9 oC, mean annual precipitation: 89–534 mm), plant productivity levels (net primary productivity (NPP): 37.9–488.3 g m⁻² yr⁻¹), soil properties (organic C: 1.1–118.3 g kg⁻¹ pH: 6.2–9.4; silt content: 5.0–51.8%; bulk density: 1.0–1.5 g cm⁻³) and microbial properties (microbial biomass C: 22.8–1101.2 mg C kg⁻¹) across the plateau (Supplementary Fig. 7; Supplementary Data 1)."

The first paragraph of the Results (lines 120-131) seems like Methods to me. There aren't any results in it. Move this paragraph to the Methods? I would.

The manuscript reads considerably better than the earlier version, as well. Thank you for taking the suggestion seriously to improve the writing.

Reviewer #3 (Remarks to the Author):

The revised version submitted by Chen et al. was well improved. I've nevertheless a major concern on the manuscript in its present form. I did not realize at the first round that carbonates were present in the soils used (in particular calcisols with some pH values around 9). Carbonates may be a CO₂ source with a $\delta^{13}\text{C}$ close the atmospheric values (~-8‰). Consequently, in the flasks, there are two sources of CO₂ in the control (SOM+carbonates) and three sources when you add labelled glucose. Using mixing equations is correct only when there are maximum two sources. Thus, you should estimate how much CO₂ came from the carbonates based on control incubation and then assuming that carbonates dynamics is similar in control flasks and in flasks with labelled glucose you can extract how came from the SOM. Without this estimation the calculations are wrong.

The study presented by Chen et al. is interesting so I suggest major revisions to let the author a chance to correct this mistake but without this correction the paper cannot be published.

Response to Reviewer #1:

[Comment 1] *Chen et al. have responded to all my previous comments, and those of the other reviewers, and I think the analysis in the paper has improved substantially. In general, the inclusion of the relative priming effects analysis has improved the confidence in the findings (but see comment below in relation to the partial correlations). By responding to reviewer 2 and including more plant variables, it appears that the main conclusion should now be that organic matter quality, as controlled by plant community composition and rates of input, determines the potential for glucose addition to induce priming effects. The slightly artificial distinction between plant variables and SOM quality is an issue that the authors address in their response to reviewer 2, when they consider how the labile C pool is controlled by the plant community. However, the fact that SOM quality is not truly independent of the plant community should be discussed more in the manuscript given the greater role plant variables play, both directly and indirectly, in the new analysis.*

[Response] Thanks for the reviewer's recognition for our revision and additional insightful comments on the revised MS. We do agree with the reviewer's comment that the SOM quality should depend on the plant variables (Jansen & Wiesenberg, 2017), **and have added one paragraph to discuss the role of plant variables in regulating SOM quality and the priming effect in the revised MS as follows:** "In addition to SOM stability, plant properties (*i.e.*, plant productivity and community composition) is another important factor regulating the priming effect, and exerted a strong indirect impact through its association with SOM stability (Fig. 4). Particularly, all the variables of SOM stability were significantly correlated with 75% of the plant variables (*e.g.*, ANPP, EVI, C input rate and the coverage of grass) (Supplementary Fig. 13). The strong dependence of SOM stability on plant properties is ascribed to the following aspects. On the one hand, as the main source of SOM, the rate of plant C inputs directly determines the amount of plant-derived C which could be newly stabilized in soil (Dungait *et al.*, 2012) and the decomposition of stable SOM pool via the priming effect (Kuzyakov, 2010). Plant C inputs can also mediate geochemical processes (*e.g.*, soil acidification) which affect soil pH and the amount of cations (Angst *et al.*, 2018), thus indirectly regulating the SOM stability. On the other hand, plant community composition controls the chemistry of plant litter and root exudates

(Hodgkins *et al.*, 2014), thereby directly affecting the chemical recalcitrance of SOM. The organic acids (*e.g.*, oxalic acid) derived from root exudates can also destabilize SOM by liberating C from association with clay minerals, thereby further increasing C accessibility (Keiluweit *et al.*, 2015). Overall, plant properties can affect the priming effect through its regulation on SOM stability.” (Page 12, Line 240-256).

Figure R1. Correlations between SOM stability and plant properties. ANPP, aboveground net primary productivity; EVI, enhanced vegetation index; Cinp, C input amount; Gra, relative coverage of grass; Sed, relative coverage of sedge; Forb, relative coverage of forb; LPI, labile pool I; LPII, labile pool II; RP, recalcitrant pool; Carb, carbohydrate; Cut, cutin-derived compound; Sub, suberin-derived compound; FeAl_d, molar ratio of free Fe/Al oxides to SOC; FeAl_o, molar ratio of amorphous Fe/Al oxides to SOC; Ca, ratio of exchangeable Ca to SOC; Mac, proportion of C occluded in macroaggregates; Mic, proportion of C protected by microaggregates; C+S, proportion of C associated with clay+silt fractions.

[Comment 2] *It is the analysis presented in Figure 2 (and Supplementary figure 4) that I think is most important. In the new version of this analysis, forbs and sedges seem to retain or increase their importance even after the direct measures of SOM stability are accounted for. Furthermore, for the SOM stability variables, when C input is included, the importance of all the SOM stability variables declines substantially. I don't think this undermines the overall conclusion of the paper but I*

think the authors do need to explain more clearly how and why they have separated the different variables into the different groups, and to discuss more how many of the SOM stability variables (especially the labile pool size and the chemical assays) are controlled by the quality and quantity of the inputs (perhaps as suggested by the structural equation modelling).

[Response] Following the reviewer's comment, we added more descriptions about the variable classification in both *Introduction* and *Methods* sessions of the revised MS as follows: "**Due to the interactive effects of SOM stability, plant, soil, and microbial properties on SOM decomposition** (Schmidt *et al.*, 2011), a comprehensive study with systematic measurements of the priming effect together with these potential drivers over a broad geographic scale is highly needed. In this study, **we quantified the relative importance of plant** (vegetation productivity and composition), **soil** (nutrient content, pH, texture and bulk density) **and microbial properties** (biomass, composition, structure, enzyme activities and stoichiometry) **and SOM stability** (chemical recalcitrance and physico-chemical protection) in regulating the priming effects based on 30 sampling sites along an approximately 2200 km grassland transect on the Tibetan Plateau" (Page 5, Line 95-105) and "Given plant, soil, and microbial properties as well as SOM stability being the major drivers of SOM decomposition and the priming effect, **to facilitate our interpretations, we classified all factors into four groups to explore their roles in regulating the priming effect: (i) plant properties** (NPP, ANPP, root biomass, enhanced vegetation index (EVI), C input amount and the relative coverage of grasses, forbs and sedges), **(ii) soil properties** (SOC content, total N content, pH, clay, silt, sand, bulk density, and soil order), **(iii) microbial properties** (bacterial PLFAs, fungi PLFAs, F/B ratio, C-, N-, P-acquiring enzyme activities, C:N ratio of enzyme activity, N:P ratio of enzyme activity, and C:P ratio of enzyme activity) **and (iv) SOM stability** (labile C pool I, labile C pool II, recalcitrant pool, carbohydrate, lignin-derived phenol, cutin-derived compounds, suberin-derived compounds, the C associated with macroaggregates, microaggregates, and the clay+silt fraction, molar $(Fe_d+Al_d)/SOC$, molar $(Fe_o+Al_o)/SOC$ and Ca_{ex}/SOC)" (Page 21, Line 425-436).

With regarding to the dependence of SOM stability variables on plant variables, we have added more descriptions in the *Results* session as follows: “However, after controlling SOM stability, the correlation coefficients between the priming intensity and plant, soil and microbial properties decreased by 33.0%, 80.1% and 97.0%, respectively. As a consequence, except for the coverage of sedge and forb, whose correlation with priming effect retained or increased, most of these factors were no longer associated with the priming effect (Fig. 2). ...In particular, the abundance of cutin-derived compounds was always significantly correlated with the priming intensity, despite its correlation with priming effect decreased by 42.9% when the glucose input amount was controlled (Supplementary Fig. 4)” (Page 8, Line 163-173).

Furthermore, as mentioned above, we added one paragraph to discuss the dependence of SOM stability on plant properties in the *Discussion* session as follows: “In addition to SOM stability, plant property (*i.e.*, plant productivity and community composition) is another important factor regulating the priming effect, and exerted a strong indirect impact through its association with SOM stability (Fig. 4). Particularly, **all the variables of SOM stability were significantly correlated with 75% of the plant variables** (*e.g.*, ANPP, EVI, C input rate and the coverage of grass) (Supplementary Fig. 13). The strong dependence of SOM stability on plant properties is ascribed to the following aspects. On the one hand, as the main source of SOM, the rate of plant C inputs directly determines the amount of plant-derived C which could be newly stabilized in soil (Dungait *et al.*, 2012) and the decomposition of stable SOM pool via the priming effect (Kuzyakov, 2010). Plant C inputs can also mediate geochemical processes (*e.g.*, soil acidification) which affect soil pH and the amount of cations (Angst *et al.*, 2018), thus indirectly regulating the SOM stability. On the other hand, plant community composition also controls the chemistry of plant litter and root exudates (Hodgkins *et al.*, 2014), thereby directly affecting the chemical recalcitrance of SOM. The organic acids (*e.g.*, oxalic acid) derived from root exudates can also destabilize SOM by liberating C from protected association with clay minerals, thereby further increasing C accessibility (Keiluweit *et al.*, 2015). Overall, plant properties can affect the priming effect through its regulation on SOM stability” (Page 12, Line 240-256).

[Comment 3] *The authors now present most of the analyses for both relative and absolute priming effects, but I think it is important that the authors also show the analysis in Figure 2 for both absolute and the relative priming effects. I apologise if I have missed this but I couldn't find the partial correlations analysis for relative priming effects. I'm also unclear why root biomass, which has been added to the plant properties does not seem to appear in Figure 2 but does appear in Supplementary Figure 4.*

[Response] Following the reviewer's comment, **we have included the root biomass into the plant properties, and also added the results of partial correlation analysis for relative priming effects in the revised MS** (Page 8, Line 173-174; Supplementary Figs. 5-6). As shown in the Figure R2, after controlling SOM stability, the correlation coefficients between the priming intensity and plant, soil and microbial properties decreased by 47.7%, 80.6% and 48.0%, respectively, despite the corresponding value for the coverage of sedge, forb and F/B ratio kept constant or even increased (Fig. R2). In contrast, the correlation coefficients decreased only by 21.4%, 22.9% and 12.7% between the priming intensity and SOM stability after removing of the controlling plant, soil and microbial properties, respectively (Fig. R3). **These results for the relative priming effects were in consistent with the results from the absolute priming effect.** We have added these descriptions in the *Results* session of the revised MS (Page 8, Line 173-174).

Figure R2. Partial correlations between the relative priming effect and the plant (a), soil (b) and microbial properties (c) after controlling SOM

stability. LPI, labile pool I; LPII, labile pool II; RP, recalcitrant pool; Carb, carbohydrate; Cut, cutin-derived compound; Sub, suberin-derived compound; FeAl_d, molar ratio of free Fe/Al oxides to SOC; FeAl_o, molar ratio of amorphous Fe/Al oxides to SOC; Ca, ratio of exchangeable Ca to SOC; Mac, proportion of C occluded in macroaggregates; Mic, proportion of C protected by microaggregates; C+S, proportion of C associated with clay+silt fractions. ANPP, aboveground net primary productivity; Root, root biomass; EVI, Enhanced Vegetation Index; C_{inp}, C input amount; Gra, relative coverage of grass; Sed, relative coverage of sedge; Forb, relative coverage of forb; SOC, soil organic carbon content; BD, bulk density; PLFA, total PLFAs; Fun, fungal PLFAs; Bat, bacterial PLFAs; F/B, fungi/bacteria ratio; C_{enz}, C-acquire enzyme activity; N_{enz}, N-acquire enzyme activity; P_{enz}, P-acquire enzyme activity; CN_{eny}, C:N ratio of enzyme activity; NP_{enz}, N:P ratio of enzyme activity; CP_{enz}, C:P ratio of enzyme activity. * $p < 0.05$, ** $p < 0.01$.

Figure R3. Partial correlations between the relative priming effect and soil organic matter (SOM) stability after controlling plant, soil and microbial properties. * $p < 0.05$, ** $p < 0.01$.

[Comment 4] *Overall, the analysis appears more robust and the dataset is very impressive. The SOM stability variables still appear to be the most important determinants of the glucose-induced priming effects, but I think the role of plant community composition and C input rate in determining these SOM stability variables needs to be emphasised more strongly.*

[Response] Thanks for the reviewer's positive comments. As mentioned above, we have emphasized the role of the plant community composition and C input rate in determining the SOM stability by adding the description of role of plant factors in the *Results* session and one-paragraph's discussion in the revised MS (Page 8, Line 165-166; Line 171-173; Page 12, Line 240-256).

[Comment 5] *Line 5: I think it needs to be clearer what the soil properties are that are not related to chemical recalcitrance or physico-chemical protection. This is not really made explicitly clear until the methods section on line 510. The distinction needs to be made earlier.*

[Response] Following the reviewer's suggestion, **to distinguish soil properties and chemical recalcitrance or physico-chemical protection, we have clearly described four groups of potential factors in the Introduction session of the revised MS as follows:** "In this study, we quantified the relative importance of plant (vegetation productivity and composition), soil (nutrient content, pH, texture and bulk density) and microbial properties (total biomass, composition, structure, enzyme activities and stoichiometry) and SOM stability (chemical recalcitrance and physico-chemical protection) in regulating the priming effects based on 30 sites along an approximately 2200 km grassland transect on the Tibetan Plateau" (Page 5, Line 100-105).

[Comment 6] *Line 187 to 189: I recognise that a proof reading service has been used but there are still a few poorly phrased sentences and this sentence contains some typos. Should it be 'indirect effects'?*

[Response] Done as suggested. We have also carefully checked the grammar throughout the revised MS.

[Comment 7] *Line 207:208: It would be helpful to explain why high SOM recalcitrance would trigger greater microbial N demand.*

[Response] Done as suggested. The sentence has been rephrased as follows: “On the other hand, high SOM recalcitrance can also trigger great microbial N demand, since the complex molecular structure can hinder the breakdown of N-containing polymers (e.g., chitin) to access available N (Langley *et al.*, 2006)”.

[Comment 8] *Lines 234-241: Despite the growing recognition of the role physico-chemical protection plays in controlling soil C storage in soils from all regions, I think it is probably still fair to suggest that C-rich soils in cool regions may have greater amounts of unprotected C than soils in other regions. Therefore, I wonder whether the authors need to consider the extent to which the findings would apply to soils in warmer climates.*

[Response] Very good comment! We agree that the role of physico-chemical protection in controlling the soil C storage may differ among various climate zones (Fang *et al.*, 2019). The results observed in this study may thus not be simply applied to other climate zones. To further advance our understanding on this issue, more studies are needed to cover various climate zones (e.g., tropical, subtropical, temperate, and alpine zone, etc). We have clearly stated this point in the revised MS as follows: “In addition, the limited ecosystem types and **climate zones** involved in this study may have also induced uncertainties for generalizing patterns and drivers observed in this study. More empirical studies with diverse ecosystem types (forests, shrubs, etc.) **and climate zones (tropical, temperate zone, etc.)** are thus needed to further advance our understanding of this issue.” (Page 13, Line 270-275).

Taken together, we appreciate for the reviewer’s insightful comments. These additional comments enabled us to have a deeper thinking on the dependence of SOM

stability on plant properties. By addressing these comments, we feel that the revised MS has been further improved. Thank you!

Response to Reviewer #2:

[Comment 1] *The reviewers have done considerable work in response to my comments and to those of the other two reviewers. Given their effort, I think the paper should be acceptable in Nature Communications. Found below are a few final suggestions for improvement. The one item where I think more description is still needed is in the selection of sites (see next point, as well). The paper states that the sites were selected for purposes of the gradient. How so, though, and for which gradients? All of the variables equally (see immediately below)? I'm only asking for more specifics about the process for site selection.*

[Response] Thanks for the reviewer's recognition for our revision and additional insightful comments on the revised MS. Regarding the site selection, we would like to mention that the gradient is not equal for all the variables. The major considerations for selecting these sampling sites are based on the precipitation gradient, because precipitation drives regional variations in vegetation (Yang *et al.*, 2009) and soil properties (Yang *et al.*, 2008) across our study area, which further determines the pattern of soil microbial community (Chen *et al.*, 2016). Therefore, as long as our sampling sites cover the precipitation gradient, we can basically cover the gradients of plant, soil and microbial properties. Nevertheless, due to the poor traffic conditions on this highest plateau around the world, we are unable to conduct systematic sampling in this area. As done by many other research groups (*e.g.*, Mu *et al.*, 2016, Li *et al.*, 2018), the sampling sites were then set along the major road at intervals of a certain distance (Ding *et al.*, 2016). We have added these detailed processes involved in site selection in the revised MS as follows: **“These sites were set throughout the geographical extent of alpine grasslands along the major road at intervals of a certain distance** (Ding *et al.*, 2016), covering a wide range of climates (mean annual temperature: -3.7–6.9 °C, mean annual precipitation: 89–534 mm), plant productivity levels (net primary productivity (NPP): 38–488 g m⁻² yr⁻¹), soil properties (organic C: 1.1–118.3 g kg⁻¹; pH: 6.2–9.4; silt content: 5–52%; bulk density: 1.0–1.5 g cm⁻³) and microbial properties (microbial biomass C: 23–1101 mg C kg⁻¹) across the plateau. **The major considerations for selecting these sampling sites were based on the precipitation gradient, because precipitation induced regional variations in vegetation (Yang *et al.*, 2009) and soil properties (Yang *et al.*, 2008) across our**

study area, which further determined the pattern of soil microbial community (Chen et al., 2016)”(Page 15, Line 310-319). Thanks for your understanding!

[Comment 2] *I think the manuscript would also be easier to grasp if all of the information about the gradient wasn't in a Supplementary table: “These sites covered a wide range of climate (Supplementary Table 1), vegetation productivity, soil and microbial properties (see details in the methods, Supplementary Data 1). Can't the authors include even a brief description of the gradients in this paragraph (lines 105-107)? This sentence from the Supplement, for instance, would be an improvement (lines 293-299): “These sites were set throughout the geographical extent of alpine grasslands, covering a wide range of climates (mean annual temperature: -3.7–6.9 oC, mean annual precipitation: 89–534 mm), plant productivity levels (net primary productivity (NPP): 37.9–488.3 g m⁻² yr⁻¹), soil properties (organic C: 1.1–118.3 g kg⁻¹ pH: 6.2–9.4; silt content: 5.0–51.8%; bulk density: 1.0–1.5 g cm⁻³) and microbial properties (microbial biomass C: 22.8–1101.2 mg C kg⁻¹) across the plateau (Supplementary Fig. 7; Supplementary Data 1).”*

[Response] Following the reviewer's suggestion, we have added more information about the gradient in the *Introduction* session as follows: “These sites covered a wide precipitation gradient, from 89 mm in arid climates up to 534 mm in humid climates. Across this precipitation gradient, net primary productivity (NPP) varied between 38 and 488 g m⁻² yr⁻¹. Both soil and microbial properties were also highly variable, with soil organic C (SOC) and microbial biomass C ranging from 1.1 to 118 g kg⁻¹ and 23 to 1101 mg C kg⁻¹, respectively (Supplementary Data 1)” (Page 5, Line 105-110).

[Comment 3] *The first paragraph of the Results (lines 120-131) seems like Methods to me. There aren't any results in it. Move this paragraph to the Methods? I would.*

[Response] We do agree with the reviewer that this paragraph seems like *Methods*, but we feel that method information is more or less needed to make the *Results* section easily understood by the readers, since *Nature Communications* requires to describe the *Methods* session at the end of the manuscript. Nevertheless, **to consider**

the reviewer's comments and also improve the readability, we have shortened this paragraph to one sentence as follows: “Both chemical recalcitrance (*i.e.*, the proportion of the labile and recalcitrant SOM factions) and physico-chemical protection (*i.e.*, the proportion of C protected by aggregates, Fe and Al oxides and exchangeable calcium) were determined to characterize the SOM stability, and then used to explore their effects on priming intensity.” (Page 6, Line 124-127). Thanks for your understanding!

Taken together, we appreciate for the reviewer's insightful comments. These additional comments enabled us to clearly describe the detailed processes about the site selection. By addressing these comments, we feel that the revised MS has been further improved. Thank you!

Response to Reviewer #3:

[Comment 1] *The revised version submitted by Chen et al. was well improved. I've nevertheless a major concern on the manuscript in its present form. I did not realize at the first round that carbonates were present in the soils used (in particular calcisols with some pH values around 9). Carbonates may be a CO₂ source with a $\delta^{13}\text{C}$ close the atmospheric values (~-8‰). Consequently, in the flasks, there are two sources of CO₂ in the control (SOM+carbonates) and three sources when you add labelled glucose. Using mixing equations is correct only when there are maximum two sources. Thus, you should estimate how much CO₂ came from the carbonates based on control incubation and then assuming that carbonates dynamics is similar in control flasks and in flasks with labelled glucose you can extract how came from the SOM. Without this estimation the calculations are wrong.*

[Response] Thanks for the reviewer's recognition for our revision and additional insightful comments on the revised MS. Following the reviewer's suggestion, we first used the mixing model to quantify the proportion of CO₂ from carbonate (f_{SIC}) in the control for 19 sampling sites containing carbonates, and found that **the CO₂ release from carbonates accounted for only 4.3% of the total CO₂ release in the control (Fig. R4)**. We then re-calculated the priming effect and found that **the corrected priming effect was highly correlated with the original priming effect** ($R^2= 0.93$, $p < 0.01$; Fig. R5). Based on these updated data, we further re-conducted the multivariate analysis and confirmed that **the overall conclusion of the study** (*i.e.*, SOM stability explained more variance in priming effect than plant, soil and microbial properties) **still held true** (Fig. R6).

To further explore whether and how much carbonates can be released as CO₂ in our study, we determined the changes in the carbonate content and its $\delta^{13}\text{C}$ value before and after the 65-day incubation. Our results showed that **there were no significant differences in both carbonate content and the corresponding $\delta^{13}\text{C}$ value before and after the incubation** (Fig. R7). Additionally, **we also conducted two NEW carbonate addition experiments** (Experiment 1: quartz sand with CaCO₃ addition; Experiment 2: ¹³C-labelled CaCO₃ addition to the one typical soil without carbonate) to quantify the contribution of carbonates to the CO₂ flux. Given that the average

inorganic C content was around 8 mg g⁻¹ soil for our soil samples (Supplementary Data), the rate of inorganic C addition was set to 8 mg g⁻¹ soil (~66.7 mg CaCO₃ g⁻¹ soil). The additional experiments showed that **CaCO₃ addition did not significantly affect the cumulative CO₂ release during the 30-day incubation in both experiments** (Fig. R8). Specifically, the ¹³C-labelled CaCO₃ experiment further revealed that **CaCO₃-derived CO₂ only accounted for 1.0% of the total CO₂ release**. Taken together, all these additional analyses and experimental results emphasized the minor contribution of carbonates to the CO₂ release from soils in this study and also its δ¹³C.

Overall, we appreciate for the reviewer's insightful comment which enabled us to have a deeper thinking on potential contributions of carbonates on CO₂ release. **To address the reviewer's comment, we performed new analyses, conducted two NEW experiments, and further discussed the potential carbonate effects by incorporating results from these additional analyses and new experiments in the revised MS as follows:** "To account for the potential effects of carbonates on CO₂ production and its δ¹³C, we determined the changes in the carbonate content and its δ¹³C value before and after the 65-day incubation and also conducted two carbonate addition experiments (Supplementary Methods). The inorganic C content and its isotope value were determined with a solid TOC analyzer (multi EA 4000, Analytik Jena, Germany) and an isotopic ratio mass spectrometry (Thermo Scientific 253 Plus, USA), respectively. Our analyses showed that there were no significant changes in either carbonate content or the corresponding δ¹³C value before and after the 65-day incubation (Supplementary Fig. 13). Moreover, CaCO₃ addition did not significantly affect the cumulative CO₂ release during the 30-day incubation in neither carbonate addition experiments (Supplementary Fig. 14). Specifically, the ¹³C-labelled CaCO₃ experiment revealed that CaCO₃-derived CO₂ only accounted for 1.0% of the total CO₂ release. These results demonstrated a minor contribution of carbonates to the CO₂ production and also its δ¹³C. Therefore, as done in most priming studies on soil with pH > 7.0 (Bastida *et al.*, 2019, Perveen *et al.*, 2019), the effects of soil carbonates on CO₂ release were considered limited in this study." (Page 18, Line 369-383). **Notably, cautions should be taken to use the corrected data about the priming effect because the correction is based on multiple assumptions (i.e.,**

carbonate dynamics is similar in control flasks and in flasks with labelled glucose; the effect of CO₂ release from carbonates on isotope value is negligible). **Moreover, as mentioned above, multiple lines of evidence demonstrated that the effects of carbonate on the CO₂ production and also its δ¹³C were relatively minor in our study. Further, this carbonate contribution was not considered in the most priming studies on soil with pH > 7.0 (Bastida *et al.*, 2019, Perveen *et al.*, 2019).** Due to the above-mentioned considerations, we prefer not to use the corrected priming effect in the revised MS. **Nevertheless, we can add the corrected results in the next round if the reviewer persists his/her opinion.** Thanks for your understanding!

Again, we really appreciate for the reviewer's insightful comment. This additional comment enabled us to have a deeper understanding on the potential contributions of soil carbonates in CO₂ release. By addressing this comment, we feel that the revised MS has been further improved. Thank you!

Figure R4. The soil organic carbon (SIC) content (a), δ¹³C value (b) and its contribution to the CO₂ production (f_{SIC}) in the control treatment (c). The δ¹³C value and f_{SIC} were determined and calculated for 19 sampling sites containing carbonates.

Figure R5. Relationship of the priming effect before and after carbonates correction, using the approach proposed by the reviewer. ***, $p < 0.001$

Figure R6. Relative contributions of SOM stability, plant, soil, and microbial properties in driving priming effect (a) and relative priming effect (b) after carbonates correction using the approach proposed by the reviewer.

Figure R7. Changes in soil inorganic carbon content (SIC, a) and its $\delta^{13}\text{C}$ value (b) before and after the experiment. G, glucose addition treatment. ns, no significant difference at the 0.05 level.

Figure R8. Effects of CaCO_3 addition on cumulative CO_2 release in new experiment 1 (EXP 1) and experiment 2 (EXP 2). The controls in EXP 1 and EXP 2 are the quartz sand and soil sample without CaCO_3 addition, respectively. ns, no significant difference at the 0.05 level.

References

- Angst G, Mueller KE, Eissenstat DM *et al.* (2018) Soil organic carbon stability in forests: Distinct effects of tree species identity and traits. *Global Change Biology*, **25**, 1529-1546.
- Bastida F, Garcia C, Fierer N *et al.* (2019) Global ecological predictors of the soil priming effect. *Nature Communications*, **10**, 3481.
- Chen YL, Ding JZ, Peng YF *et al.* (2016) Patterns and drivers of soil microbial communities in Tibetan alpine and global terrestrial ecosystems. *Journal of Biogeography*, **43**, 2027-2039.
- Ding JZ, Li F, Yang GB *et al.* (2016) The permafrost carbon inventory on the Tibetan Plateau: a new evaluation using deep sediment cores. *Global Change Biology*, **22**, 2688-2701.
- Dungait JaJ, Hopkins DW, Gregory AS, Whitmore AP (2012) Soil organic matter turnover is governed by accessibility not recalcitrance. *Global Change Biology*, **18**, 1781-1796.
- Fang K, Qin S, Chen L, Zhang Q, Yang Y (2019) Al/Fe mineral controls on soil organic carbon stock across Tibetan alpine grasslands. *Journal of Geophysical Research: Biogeosciences*, 10.1029/2018JG004782.
- Hodgkins SB, Tfaily MM, Mccalley CK *et al.* (2014) Changes in peat chemistry associated with permafrost thaw increase greenhouse gas production. *Proceedings of the National Academy of Sciences*, **111**, 5819-5824.
- Jansen B, Wiesenberg GLB (2017) Opportunities and limitations related to the application of plant-derived lipid molecular proxies in soil science. *Soil*, **3**, 211-234.
- Keiluweit M, Bougoure JJ, Nico PS, Pett-Ridge J, Weber PK, Kleber M (2015) Mineral protection of soil carbon counteracted by root exudates. *Nature Climate Change*, **5**, 588-595.
- Kuzyakov Y (2010) Priming effects: Interactions between living and dead organic matter. *Soil Biology & Biochemistry*, **42**, 1363-1371.
- Langley JA, Chapman SK, Hungate BA (2006) Ectomycorrhizal colonization slows root decomposition: the post-mortem fungal legacy. *Ecology Letters*, **9**, 955-959.
- Li J, Yan D, Pendall E *et al.* (2018) Depth dependence of soil carbon temperature sensitivity across Tibetan permafrost regions. *Soil Biology and Biochemistry*, **126**, 82-90.
- Mu CC, Zhang TJ, Zhao Q, Guo H, Zhong W, Su H, Wu QB (2016) Soil organic carbon stabilization by iron in permafrost regions of the Qinghai-Tibet Plateau. *Geophysical Research Letters*, **43**, 10286-10294.
- Perveen N, Barot S, Maire V *et al.* (2019) Universality of priming effect: An analysis using thirty five soils with contrasted properties sampled from five continents. *Soil Biology and Biochemistry*, **134**, 162-171.
- Yang Y, Fang J, Tang Y, Ji C, Zheng C, He J, Zhu B (2008) Storage, patterns and controls of soil organic carbon in the Tibetan grasslands. *Global Change Biology*, **14**, 1592-1599.
- Yang YH, Fang JY, Pan YD, Ji CJ (2009) Aboveground biomass in Tibetan grasslands. *Journal of Arid Environments*, **73**, 91-95.

REVIEWERS' COMMENTS:

Reviewer #1 (Remarks to the Author):

The manuscript has been improved substantially and I think represents a very clear and balanced interpretation of the results, and a novel and important contribution on the priming of SOM.

I would recommend that this paper is now acceptable for publication in Nature Communications and thank the authors for dealing with my previous comments so thoroughly.

Iain Hartley

Reviewer #3 (Remarks to the Author):

All the referees comments were not easy to tackle and the authors did a great job to answer to all the points. I suggest to accept the manuscript.

Response to Reviewer #1:

[Comment] The manuscript has been improved substantially and I think represents a very clear and balanced interpretation of the results, and a novel and important contribution on the priming of SOM.

I would recommend that this paper is now acceptable for publication in Nature Communications and thank the authors for dealing with my previous comments so thoroughly.

[Response] Thanks for the reviewer's recognition for our revision.

Response to Reviewer #3:

[Comment] All the referees comments were not easy to tackle and the authors did a great job to answer to all the points. I suggest to accept the manuscript.

[Response] Thanks for the reviewer's recognition for our revision.